# GriDiT: Factorized Grid-Based Diffusion for Efficient Long Image Sequence Generation

**Snehal Singh Tomar**[*], **Alexandros Graikos, Arjun Krishna, Dimitris Samaras, Klaus Mueller**[*]

*Department of Computer Science*
*Stony Brook University, NY, USA*
[*]*Correspondence: stomar@cs.stonybrook.edu, mueller@cs.stonybrook.edu*

**Reviewed on OpenReview:** *https://openreview.net/forum?id=QLD47Ou5lp*

## Abstract

Modern deep learning methods typically treat image sequences as large tensors of sequentially stacked frames. However, is this straightforward representation ideal given the current state-of-the-art (SoTA)? In this work, we address this question in the context of generative models and aim to devise a more effective way of modeling image sequence data. Observing the inefficiencies and bottlenecks of current SoTA image sequence generation methods, we showcase that rather than working with large tensors, we can improve the generation process by *factorizing* it into first generating the *coarse sequence* at low resolution and then refining the *individual frames* at high resolution. We train a generative model solely on *grid images* comprising subsampled frames. Yet, we learn to generate *image sequences*, using the strong self-attention mechanism of the Diffusion Transformer (DiT) to capture correlations between frames. In effect, our formulation extends a 2D image generator to operate as a low-resolution 3D image-sequence generator without introducing any architectural modifications. Subsequently, we super-resolve each frame individually to add the sequence-independent high-resolution details. This approach offers several advantages and can overcome key limitations of the SoTA in this domain. Compared to existing image sequence generation models, our method achieves superior synthesis quality and improved coherence across sequences. It also delivers high-fidelity generation of arbitrary-length sequences and increased efficiency in inference time and training data usage. Furthermore, our straightforward formulation enables our method to generalize effectively across diverse data domains, which typically require additional priors and supervision to model in a generative context. Our method consistently delivers superior quality and offers a $> 2\times$ speedup in inference rates across various datasets. Code is available at our project page.

## 1 Introduction

Image sequences form some of the richest perceptual signals in nature, constituting the largest volume of shared data on the internet. At the same time, the complexity of their many degrees of variability (lighting, motion, camera effects, etc.) makes them hard to model. These difficulties have significantly impeded advancements in image sequence generative models, with image sequence generation lagging significantly behind the image and natural language generation paradigms. Only recently has there been an interest (Cai et al., 2024; Dalal et al., 2025; Bian et al., 2025) in exploring self-attention-driven architectures (Vaswani et al., 2017; Dosovitskiy et al., 2021; Peebles & Xie, 2023) for conditional video generation. However, the benefits of these architectures have yet to be fully exploited by the field.

In the context of this work, we consider *image sequence generation* as an umbrella task encompassing video generation, where the sequential ordering need not necessarily be temporal (e.g., Computed Tomography (CT) scans). Several meaningful attempts have been made to perform the task in the past. However, these attempts fail to bridge long-standing research gaps.

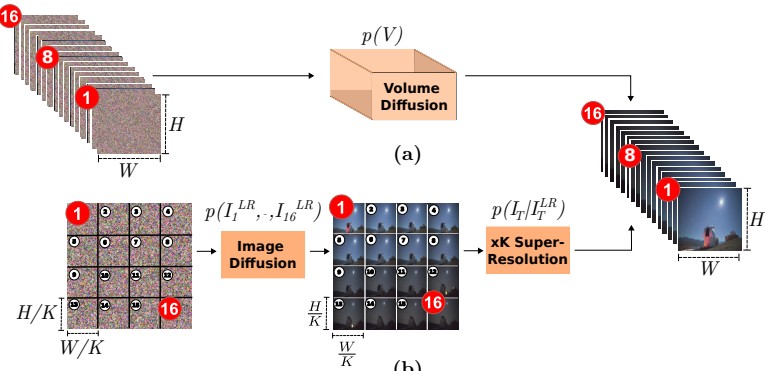

Figure 1: **(a)** SoTA image sequence generation models treat image sequences as large tensors of ordered frames. In contrast, **(b)** our method factorizes image sequence generation into two stages. First, we learn to model the dynamics of the sequence at low resolution, treating the frames as subsampled image grids. Second, we learn to super-resolve individual frames at high resolution. Using the DiT's self-attention mechanism to model dynamics across frames, and paired with our sampling strategy, our method yields superior synthesis quality for sequences of arbitrary length while significantly reducing sampling time and training data requirements. (Notation used is the same as defined in section 3. $K = 4$.)

We envision that the solution to efficient synthesis of arbitrary-length image sequences lies in a well-formed recipe that combines a more effective data representation, an efficient model architecture, and autoregressive sampling. To that effect, we introduce **GriDiT**: We represent image sequences as grid images comprising subsampled frames, which enables us to factorize image sequence generation into efficient coarse sequence generation at low resolution, followed by refinement of individual frames at high resolution. We pair our generation pipeline with our proposed *Grid-based Autoregressive Sampling Algorithm* to sample sequences of arbitrary length. Figure 1 illustrates our approach and contrasts it with prior work.

Our extensive experiments on the SkyTimelapse (Zhang et al., 2020), CT-RATE (Hamamci et al., 2024a;b), Minecraft (Yan et al., 2023), and Taichi (Siarohin et al., 2019) datasets demonstrate that our approach surpasses current state-of-the-art methods in long-range consistency, frame-wise quality, and sampling efficiency for image sequence generation. These advantages become increasingly evident as video length grows. Ablation studies reveal that the core of our improvement lies in coupling Grid-based Modeling with 3D positional embeddings to harness the DiT's (Peebles & Xie, 2023) self-attention mechanism effectively. We summarize the key contributions of our work as:

- We present a pragmatic outlook towards generating image sequences. We factorize the process into generating coarse sequences at low resolution (Stage 1) and refining individual frames at high resolution (Stage 2). Our approach departs from conventional sequence modeling by treating sequences as image grids, thereby allowing us to use an *image* generation model for *image-sequence* generation.

- We leverage the Diffusion Transformer's self-attention mechanism with our 3D positional embeddings to ensure long-range consistency between the generated frames, achieving superior perceptual quality than the current SoTA on several datasets.

- The proposed efficient modeling approach surpasses SoTA methods in sampling time ($> 2\times$ faster), training data required (similar performance with 10% of the training data in data-critical domains), and simplicity of architectures, enabling its applicability to challenging data domains without a domain-specific design.

- We facilitate SoTA, arbitrarily long (up to 1024 frames), frame roll out for image sequence generation by introducing a Grid-based Autoregressive sampling algorithm for our diffusion model.

## 2 Prior Work

The image-sequence generation domain has primarily been driven by an *efficiency* versus *synthesis quality* trade-off in recent years. Methods have attempted to make the problem tractable by either modifying model architectures and optimization objectives or utilizing superior embeddings of the sequence. Our position in this work is entirely different from prior art. We present an alternative way of *looking at the data itself* and harness it to tackle the long-existing trade-off effectively.

The high computational cost of processing large video tensors has been a significant impediment in the advancement of image-sequence generation (He et al., 2022; Yan et al., 2021; Tulyakov et al., 2018; Tian et al., 2021; Yu et al., 2023; Skorokhodov et al., 2022; Yu et al., 2022; Park et al., 2024; Brooks et al., 2022; Girdhar et al., 2025; Guo et al., 2024). Most methods (Yan et al., 2021; Tulyakov et al., 2018; Tian et al., 2021; Brooks et al., 2022; Blattmann et al., 2023; Girdhar et al., 2025) model videos as large tensors, which limits the maximum sequence length and incurs slow inference rates. Recent works using DiTs (Cai et al., 2024; Bian et al., 2025; Dalal et al., 2025) focus on architectural improvements for better conditioning; we consider these concurrent, but orthogonal to our goal of rethinking image sequence modeling. Approaches leveraging proxy models such as Implicit Neural Representations (INRs) (Skorokhodov et al., 2022; Yu et al., 2022; Park et al., 2024) trade off perceptual quality for efficiency. Factorized generation has shown promise: Emu Video and AnimateDiff (Girdhar et al., 2025; Guo et al., 2024) split text-to-video into text-to-image and image-to-video stages. In contrast, LongVideoGAN (Brooks et al., 2022) factorizes within a Generative Adversarial Network (GAN) based framework but lacks support for arbitrary-length sequences or resolutions beyond 256×256. There is a paucity of methods (Skorokhodov et al., 2022; He et al., 2022; Yu et al., 2023; Ge et al., 2022) that attempt to generate arbitrary-length videos. Of these, PVDM (Yu et al., 2023) and LVDM (He et al., 2022) are latent diffusion-based approaches. Whereas TATS (Ge et al., 2022) utilizes a GAN, StyleGAN-V is a GAN approach paired with INRs. We compare our method with all relevant techniques to ensure coverage of the various approaches taken to solve the problem. For general image sequence synthesis, we consider the generation of 3D CT volumes. In this regard, GenerateCT (Hamamci et al., 2024c) is the only method that reports spatio-temporal consistency metrics on publicly available 3D CT data, making it our primary baseline.

A recent line of work (Chen et al., 2025; Ruhe et al., 2024) explores alternate noising schemes for the task. Of these, we compare with Diffusion Forcing (Chen et al., 2025), which forms a pertinent baseline for comparison as it utilizes autoregressive sampling and its implementation is publicly available. We defer further commentary and contrast with prior art along with a detailed review of the applications of relevant multiscale approaches in generative modeling to an exhaustive related work section in Appendix A.1. To the best of our knowledge, no prior approach in the domain has viewed the problem of image-sequence generation from a standpoint akin to ours.

## 3 Our Method

**Overview.** As shown in Figure 2, we start by modeling 3D image sequences as 2D image grids comprising subsampled frames while preserving their sequential ordering. We then proceed to train an unconditional DiT (Peebles & Xie, 2023) (Stage 1 (①)) on these images using the standard DDPM (Ho et al., 2020) training procedure. At this stage, our model learns to synthesize coarse low-resolution image sequences in the form of 2D grid images. Subsequently, we utilize a conditional DiT-based Super Resolution (SR) (Stage 2 (②)) pipeline as an up-sampling and refinement mechanism for the generated low-resolution image sequence elements, which we first extract from their respective grids in an order-preserving manner to synthesize high-resolution 3D image sequences. Finally, for inference, we introduce *Grid-based Autoregressive sampling*, which allows us to build on the learned DiT's self-attention mechanism to sample arbitrary-length sequences while only having learned to generate 2D, RGB grid images. We provide a brief preliminary on DDPMs in Appendix A.2 for completeness. Both the employed models, viz. Stage-1 and Stage-2 are latent diffusion models trained in the Stable Diffusion (Rombach et al., 2022) Variational Autoencoder's (VAE's) latent space.

**Notation.** We follow the Notation described here consistently throughout this work. **Data: V** denotes an image sequence and **I** denotes an image in the sequence. $\bar{N}, T, H, W, L$, and $K$ denote the number of

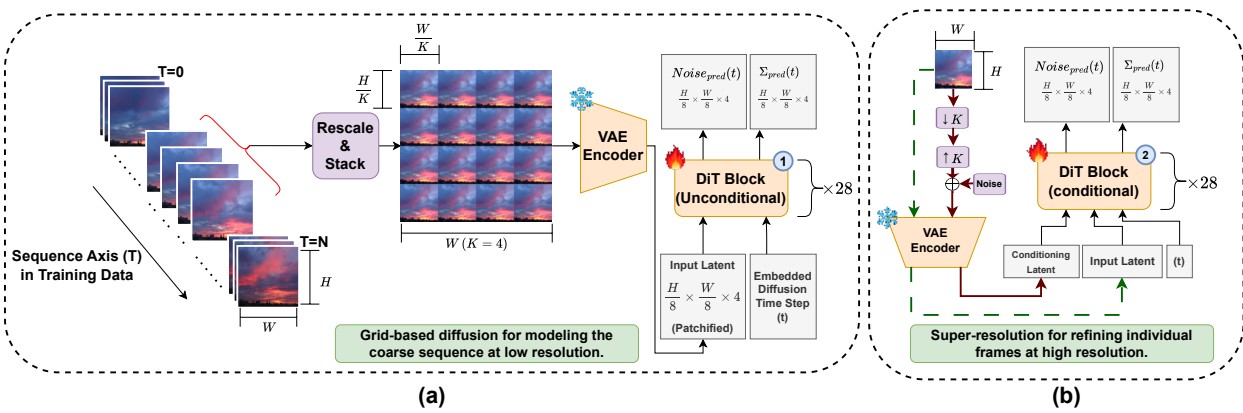

**(a)**     **(b)**

Figure 2: An overview of our method's training pipeline. We leverage DiT's self-attention for efficient, high-quality, and arbitrary-length image sequence generation using a two-stage process. **(a) Stage 1** (①): We learn to generate the *coarse image sequence* at low resolution. We organize the image sequence as grid images, comprising subsampled frames arranged in their sequential order. An unconditional latent DiT is trained to generate them. **(b) Stage 2** (②): We learn to refine *individual frames* in the generated coarse sequence via faithful generative $\times K$ super-resolution. We pose the problem as one of learning a conditional DiT model to restore the degradation caused by the lossy subsampling of images from our training dataset. (🔥: trainable. ❄: frozen during training. $\downarrow K$ & $\uparrow K$: bilinear (lossy) downsampling and upsampling, respectively. The "Noise" function is further elaborated upon in section 3.3.)

frames in a training image sequence, frame index, frame height, frame width, the total length of the synthetic sequence, and the number of rows and columns in the extracted grid image, respectively. **Diffusion:** $t, \mathbf{X}_t, \mathcal{T}, \mathcal{T}_s, \alpha_t, \bar{\alpha}_t, \epsilon, \epsilon_\theta(\mathbf{X}_t, t), \beta_t, \theta, \mathcal{N}(0, \boldsymbol{I})$, and $\sigma_t$ denote the diffusion timestep, sample at timestep $t$, total number of diffusion timesteps used in training, total number of diffusion timesteps used in sampling, noise mean coefficient, cumulative noise mean coefficient, noise sampled from standard normal distribution, noise predicted by a DiT model, $1 - \alpha_t$, model parameters, standard normal distribution, and noise standard deviation coefficient at timestep $t$, respectively. **Grid-based Autoregressive Sampling:** $T'$ denotes the autoregressive sampling iteration in step 1. $T'' = i[i+1]$ denotes diffusion inpainting-based frame interpolation between the novel frames generated at $T' = i$ and $T' = i+1$ in step 2.

### 3.1 Generating Image Sequences with GriDiT

For fixed-length image sequences and videos (or more generally, 2D image volumes) of length $\bar{N}$, a simple video generative model would learn how to sample from the joint distribution of the frames

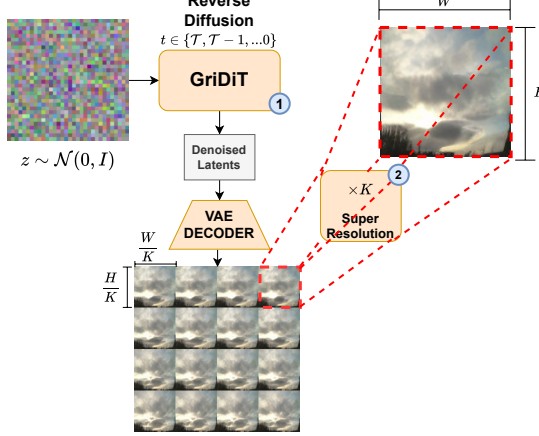

Figure 3: Inferring a single grid image's sequence elements from our model entails: **(1)** synthesizing grid images using Stage 1 (①), **(2)** splitting the grid into coarse frames, **(3)** adding fine information and super-resolving the coarse frames into individual output frames via Stage 2 (②), and **(4)** stacking the ordered frames to form the sequence.

$$p(\mathbf{V}) = p(\mathbf{I}_1, \mathbf{I}_2, \dots, \mathbf{I}_T, \dots \mathbf{I}_{\bar{N}}). \tag{1}$$

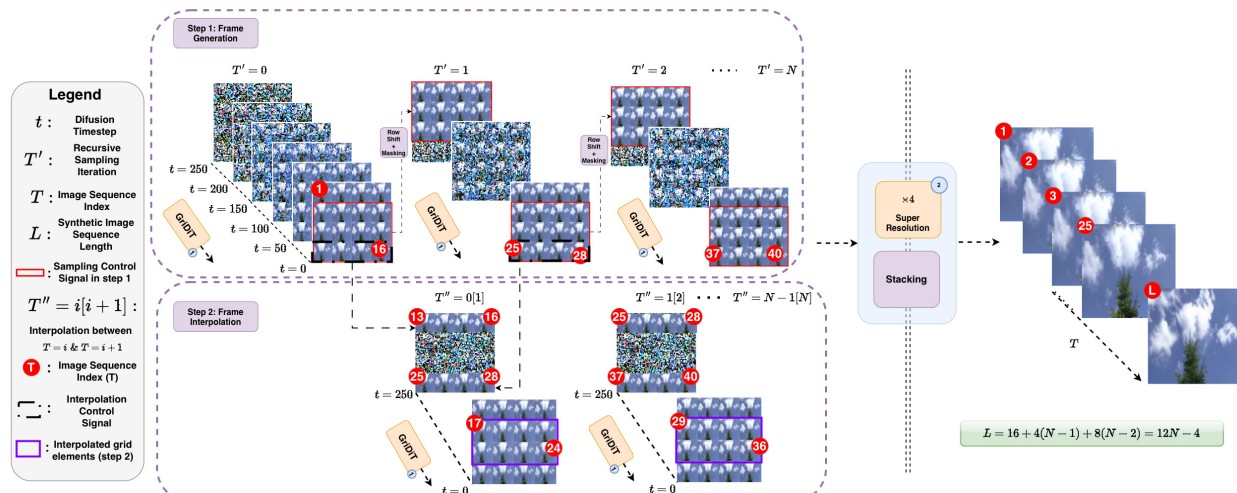

Figure 4: We illustrate our **Grid-based Autoregressive Sampling Algorithm** used to sample arbitrary-length image sequences. The algorithm entails two steps. We start with **step 1** wherein the first iteration starts with vanilla Stage 1 (①) sampling. Every subsequent iteration uses an appropriately noised control signal from the previous iteration's output at each reverse diffusion timestep to generate four new grid elements, which are in spatiotemporal agreement with all previous grid elements. Upon transitioning to **step 2**, we interpolate eight new frames between each consecutive pair of 4 new frames generated in different sampling iterations of step 1 for superior temporal resolution. Finally, all new coarse grid-elements are super-resolved via Stage 2 (②) and stacked in their sequential order for superior spatial resolution and refinement. Consequently, we obtain a long, high-quality image sequence. $N$ such iterations lead to an $L = 12N - 4$ length image sequence, inducing a substantial gain in efficiency and quality over the SoTA.

In this work, we propose an alternative way (shown in Figure 1) of learning to generate $\mathbf{V}$ by introducing latent variables that represent the low-resolution frames. Instead of modeling the joint distribution of the high-resolution frames, we model the joint distribution of low-resolution frames, which is cheaper to sample from and can adequately represent the coarse motion between frames. Then, for each frame, we also train a super-resolution module that adds motion-independent details to refine each low-res frame separately. With this factorization, we disentangle the motion and high-resolution appearance into two separate models, which we find is overall cheaper than modeling the full distribution, while also maintaining high sample quality. Using the previous notation and for $\mathbf{I}_T^{LR}$ being the low-resolution frames, we can express:

$$p(\mathbf{I}_1, \ldots, \mathbf{I}_{\bar{N}}, \mathbf{I}_1^{LR}, \ldots, \mathbf{I}_{\bar{N}}^{LR}) = p(\mathbf{I}_1, \ldots, \mathbf{I}_{\bar{N}} \mid \mathbf{I}_1^{LR}, \ldots, \mathbf{I}_{\bar{N}}^{LR}) \, p(\mathbf{I}_1^{LR}, \ldots, \mathbf{I}_{\bar{N}}^{LR})$$

$$= p(\mathbf{I}_1 \mid \mathbf{I}_1^{LR}) \cdots p(\mathbf{I}_{\bar{N}} \mid \mathbf{I}_{\bar{N}}^{LR}) \, p(\mathbf{I}_1^{LR}, \ldots, \mathbf{I}_{\bar{N}}^{LR}), \quad (2)$$

Where $p(\mathbf{I}_1^{LR}, \ldots, \mathbf{I}_{\bar{N}}^{LR})$ models the joint distribution of the low-res frames represented as grid images and $p(\mathbf{I}_T \mid \mathbf{I}_T^{LR})$ super-resolves each frame. We remark that $p(\mathbf{I}_1, \ldots, \mathbf{I}_{\bar{N}}, \mathbf{I}_1^{LR}, \ldots, \mathbf{I}_{\bar{N}}^{LR})$ is easily tractable given $p(\mathbf{V})$ (Eq. 1) because $\mathbf{I}_T^{LR}$ can be obtained from $\mathbf{I}_T$ by bicubic downsampling. We parameterize learning the learned distribution of low resolution frames represented by grid images as $p_{\theta_1}(\mathbf{I}_1^{LR}, \ldots, \mathbf{I}_{\bar{N}}^{LR})$, and denote the learned optimal stage 1 (①) parameters as $\theta_1^*$. Similarly, we parameterize learning the refinement of coarse individual frames as $p_{\theta_2}(\mathbf{I}_T \mid \mathbf{I}_T^{LR})$, and denote the learned optimal stage 2 (②) parameters as $\theta_2^*$. We elucidate our model architectures in Section A.3 of the Appendix.

Here, we emphasize that learning to model $p(\mathbf{I}_1^{LR}, \ldots, \mathbf{I}_{\bar{N}}^{LR})$ with GriDiT is significantly less compute-intensive than learning to model $p(\mathbf{V})$ (Eq 1) via SoTA methods. Hence, this formulation is crucial for the performance of our method, as highlighted in our experiments.

### 3.2 Grid-based Frame Modeling

To model sequences as grids of images comprising subsampled frames, we arrange a subsequence of $K^2$ frames, subsampled by a factor $K$, into a grid per their sequential ordering to form an image representing the frames (or slices) of the video (or volume) data, at a lower spatial resolution. The process is illustrated in Figure 2 and further expanded formally in Appendix A.4. Here, we remark that our grid-image formulation does not alter the temporal ordering in the image sequences in any way. Instead, it simply *rearranges* the (subsampled) frames per their original temporal order. The obtained grid image tensor is now suitable for training the image diffusion model. We employ an unconditional DiT model (Stage 1 (①)) (Peebles & Xie, 2023) to learn to sample from the distribution of grid images $p_{\theta_1}(\mathbf{I}_1^{LR}, \ldots, \mathbf{I}_{\tilde{N}}^{LR})$. We use $K = 4$ in most experiments.

**3D Positional Embeddings.**  The DiT model uses fixed 2D positional embeddings to inform each patch of its spatial location. Since we are modeling multiple frames in a single image, each pixel should not only be aware of its spatial neighbors, but also of the neighboring pixels in different frames. We employ 3D positional embeddings to encode the cross-frame locality. This is a straightforward extension of the 3D positional embeddings used in video transformer models (Arnab et al., 2021). We first compute the 3D positional embeddings on the downsampled image volume and rearrange them into the grid format to combine with the 2D embeddings. We present a formal description of our positional embeddings in Appendix A.6.

**Grid-based Autoregressive Sampling.**  We posit that Autoregressive sampling holds the key to generating high-quality, arbitrary-length image sequences. To that end, we introduce Grid-based Autoregressive sampling to sample arbitrary-length image sequences from our unconditional DiT (Stage 1 (①)) model, which is trained solely to generate 2D RGB grid images. These grid images comprise $K^2$ length sub-sequences in the form of subsampled grid elements. The two-step approach, illustrated by Figure 4, draws inspiration from diffusion-based image inpainting literature (Lugmayr et al., 2022). Diffusion inpainting *fills* "missing" segments of an image that are coherent with the "known" segments by modifying the reverse-diffusion (sampling) process. The modification substitutes the regions corresponding to the "known" segments within the denoised latent (starting from $\mathcal{N}(0, I)$ at $t = \mathcal{T}$) at each reverse-sampling timestep $t$ with their appropriately forward-noised variants (at $t$) obtained from the ground truth. Thereby allowing the diffusion process and the denoiser model's priors to *inpaint* "missing" information in accordance with the "known" information. Conditional diffusion inpainting of "missing" grid elements, in accordance with "known" grid elements, forms the foundation of our sampling approach. In our case, the diffusion process ensures spatial coherence, and the Stage 1 (①) model's learned implicit temporal bias from grid images ensures temporal coherence.

In Step 1 of our algorithm, we generate a coarse sequence of grid elements that maintain spatiotemporal coherence. We begin with vanilla DDPM sampling to generate the start-of-sequence grid image ($K^2$ new grid elements). Each subsequent iteration $T'$ results in the generation of 4 new coarse frames that bear spatiotemporal coherence with the 12 immediately previous frames. The coherence is ensured via conditional diffusion inpainting, with the last three rows of the prior iteration, $T' - 1$, serving as the sampling control signal. The coarse frames obtained here act as inputs to step 2.

Step 2 of our algorithm focuses on enhancing the temporal resolution of the sequence. At each iteration $T'' = [i]i + 1$, We interpolate eight new grid elements between the novel grid elements synthesized at $T' = i$ and $T' = i + 1$. We do so by using the latest synthetic row from $T' = i$ as the first row and that from $T' = i + 1$ as the last row in our conditional diffusion inpainting framework.

Finally, we split and stack all newly created grid elements according to their intended temporal order. The resulting coarse frames are now suitable for spatial super-resolution and refinement using our Stage 2 model. Upon conclusion, $N$ stage 1 sampling iterations appropriately followed by $N - 1$ step 2 iterations lead to a synthetic sequence of length $L = 12N - 4$. Here we make two critical remarks. First, we execute step 2 of our sampling scheme *after* step 1, rather than in an alternating fashion. We simply set $N$ via $N = (L + 4)/12$ where $L$ is the required number of output frames. Second, the inclusion of a second interpolation-based sampling step is motivated by the need to mitigate subtle discontinuities and improve temporal stability that persist when *only* Step 1 is applied. Beyond these qualitative gains, the two-step scheme also accelerates sampling: each Step 2 iteration generates eight new grid elements, compared to four in Step 1, thereby

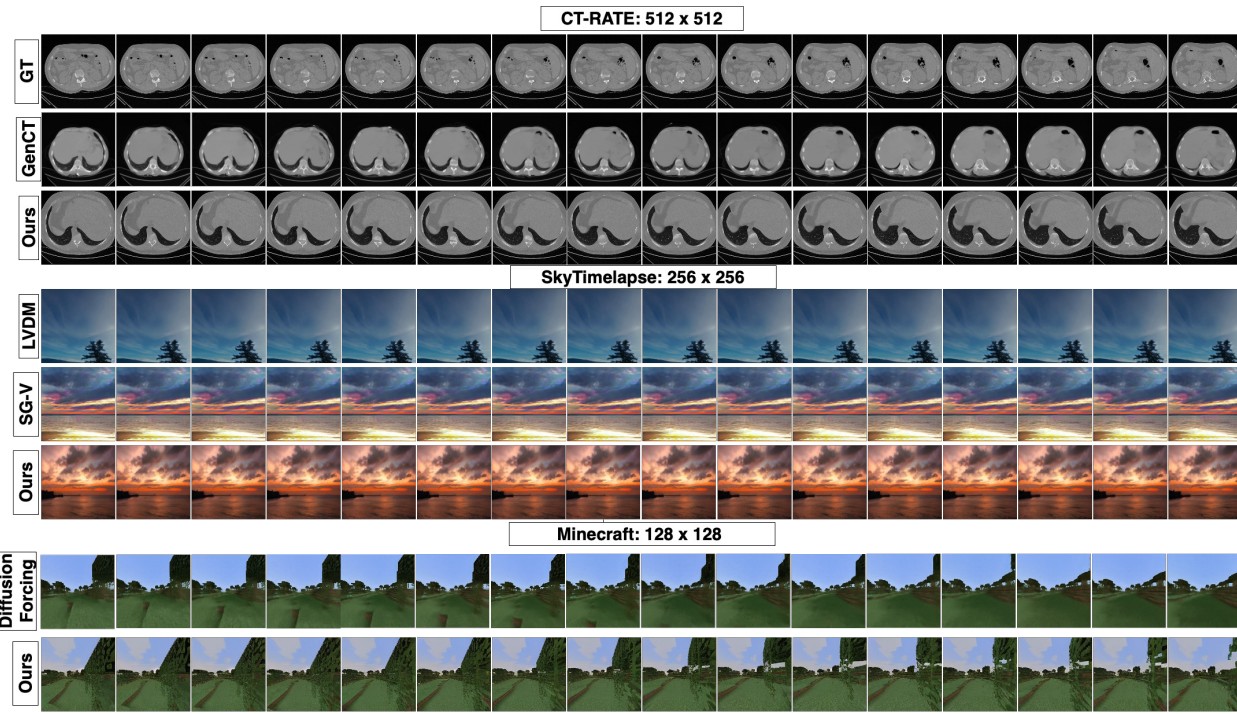

Figure 5: **Qualitative comparisons** with the SoTA on the CT-RATE SkyTimeLapse, and Minecraft datasets. Images are arranged from left to right in their sequential order, i.e., frames 1 through 16. Synthetic CT volumes are generated at $512 \times 512$ resolution using $\times 4$ SR whereas SkyTimelapse videos are generated at $256 \times 256$ resolution using $\times 2$ SR. We use the standard $4 \times 4$ grid setting in Grid-based Autoregressive sampling (step-1) for both cases. Whereas we use a $\{8 \times 8$ grid, four-row control signal$\}$ setting for step 1 of sampling in experiments on the Minecraft dataset to intentionally allow room for $\times 2$ SR, ensuring a fair comparison while yielding the desired $128 \times 128$ resolution. Our method yields superior performance in terms of spatiotemporal coherence and quality. (GT: ground truth, GenCT: GenerateCT, SG-V: StyleGAN-V)

Table 1: 3D CT Volume synthesis benchmark. **Bold:** best. $\dagger$: numbers reported by GenerateCT .

| Method | Data % | FVD ↓ | FID ↓ | Time (s) ↓ |
|---|---|---|---|---|
| Base w/ Imagen | 100 | 3557.7$^\dagger$ | 160.8$^\dagger$ | 234 |
| Base w/ SD | 100 | 3513.5$^\dagger$ | 151.7$^\dagger$ | 367 |
| Base w/ Phenaki | 100 | 1886.8$^\dagger$ | 104.3$^\dagger$ | 197 |
| GenCT | 100 | 1092.3$^\dagger$ | 55.8$^\dagger$ | 184 |
| Ours | 10 | 1089.5 | 68.2 | **53.8** |
| Ours | 60 | 1079.6 | 61.5 | **53.8** |
| **Ours** | 100 | **998.43** | 54.8 | **53.8** |

Table 2: Quantitative comparisons on the Minecraft dataset. **Bold:** best.

| Method | Number of Synthetic Frames | | | |
|---|---|---|---|---|
| | 16 | 128 | 256 | 1024 |
| | FVD ↓ | | | |
| Diffusion Forcing | 62.43 | 199.117 | 221.53 | 261.23 |
| Ours | 64.32 | **184.728** | 218.69 | **243.21** |
| | Flicker (Average $l_1$ distance between frames) ↓ | | | |
| Diffusion Forcing | **20.42** | **21.99** | 27.63 | 32.13 |
| Ours | 20.79 | 22.31 | **25.68** | **31.86** |

reducing the total number of required iterations. A formal overview of our sampling technique, along with algorithmic summaries of Steps 1 and 2, can be found in Appendix A.5.

## 3.3 Frame Refinement by Generative Super-Resolution (SR)

We use super-resolution to refine individual frames from the coarse sequence generated via Grid-based Autoregressive sampling from the learned distribution $p_{\theta_1^*}(\mathbf{I}_1^{LR}, \ldots, \mathbf{I}_{\bar{N}}^{LR})$. We specifically chose a diffusion model to perform SR, which we denote as $p_{\theta_2^*}(\mathbf{I}_T | \mathbf{I}_T^{LR})$, to hallucinate some details that are lost in the

low-resolution images. Vanilla DiT does not support direct image conditioning. Therefore, we make specific modifications in our Stage 2 (②) architecture. As illustrated in Figure 2 (b), we train stage-2 to super-resolve the low-resolution frames using a conditional DiT model. In this case, we utilize the DiT model with certain modifications, as we are generating a single, high-resolution (HR) image conditioned on its low-resolution (LR) counterpart. More specifically, we obtain LR images from our training datasets of HR images by applying a combination of degradations (successive lossy down and upsampling, noise addition, and blurring) to the HR images. We elaborate on the employed degradation scheme, providing experimental justification for it in Appendix A.8. Our training dataset for the task now comprises several {LR, HR} pairs.

For each pair, the SR model's goal is to learn to generate HR images conditioned on the corresponding LR input. We train it to do so by embedding both LR and HR images in the VAE's (Kingma & Welling, 2022) latent space, concatenating the two latents, projecting the concatenated latent on the original hidden dimension, and training the DiT to generate the embedding for HR given the obtained projected embedding as input. We apply conditioning to the DiT via adaptive layer norm. We summarize the process of Stage-2 inference, or going from a coarse synthetic grid image comprising subsampled frames to highly photorealistic and motion preserving individual frames, in Figure 3.

## 4 Experiments

### 4.1 Setup

**Datasets.** We make use of the Skytimelapse (Zhang et al., 2020), Taichi (Siarohin et al., 2019), and Minecraft (Yan et al., 2023) datasets at $256^2$, $256^2$, and $128^2$ resolution, respectively, for evaluating our method on the arbitrary length video generation task. We utilize the CT-RATE dataset (Hamamci et al., 2024a;b) at a resolution of $512^2$ for evaluations on the 3D CT Volume generation task.

**Experimental Specifics.** We use $K = 4$, and $\mathcal{T}_s = 250$ in all our experimental results except wherever we specify otherwise. We defer other design choices with respect to training and experimentation to Appendix A.7

**Baselines.** We construct relevant baselines with widely benchmarked prior works to compare against our model's performance. Our baselines encompass both GAN and diffusion-based methods for completeness. In the context of video generation: (1) On the SkyTimelapse dataset we compare with VideoGPT (Yan et al., 2021), MoCoGAN (Tulyakov et al., 2018), MoCoGAN-HD (Tian et al., 2021), LVDM (He et al., 2022), PVDM (Yu et al., 2023), DIGAN (Yu et al., 2022), StyleGAN-V (Skorokhodov et al., 2022), and DDMI (Park et al., 2024) for standard (length: 16 an 128 frames) and with StyleGAN-V (Skorokhodov et al., 2022) and LVDM (He et al., 2022) for arbitrarily long generation, respectively. Our choice of competing methods is derived from StyleGAN-V (Skorokhodov et al., 2022) and DDMI (Park et al., 2024); (2) On the Taichi dataset, we compare with LVDM (He et al., 2022), TATS (Ge et al., 2022), DIGAN (Yu et al., 2022), and Style-SV (Zhang et al., 2023) in different length settings as appropriate for each method. This choice of baselines is dictated by various other relevant methods in the domain. For image sequence (3D CT Volume) generation on the CT-RATE dataset, we borrow our baselines from GenerateCT (Hamamci et al., 2024c), which were formed by appropriately finetuning Stable Diffusion (Base w/ SD) (Rombach et al., 2022), Imagen (Base w/ Imagen) (Saharia et al., 2022), and Phenaki (Base w/ Phenaki) (Villegas et al., 2023). We note that Phenaki is a video-generation model, and comparing it ensures completeness with respect to the types of chosen competing methods. We compare with only the best variants of competing methods. For the sake of fairness in comparison, we borrowed metrics reported in prior work and utilized publicly available pre-trained weights wherever feasible. We retrain certain modules of other methods if they are fit for comparison on the experiment in question and do not report weights or provide pre-trained weights. More specifically, we retrained relevant modules of LVDM and Style-SV on the Taichi dataset. In Appendix A.1.1, we present our concurrent works and define the scope of our comparative analysis.

### 4.2 Our Results

**Synthesis quality of arbitrary length sequences.** Conventional fixed-length image sequence synthesis models treat videos as stacked frame tensors, inherently limiting sequence-length flexibility. Consequently,

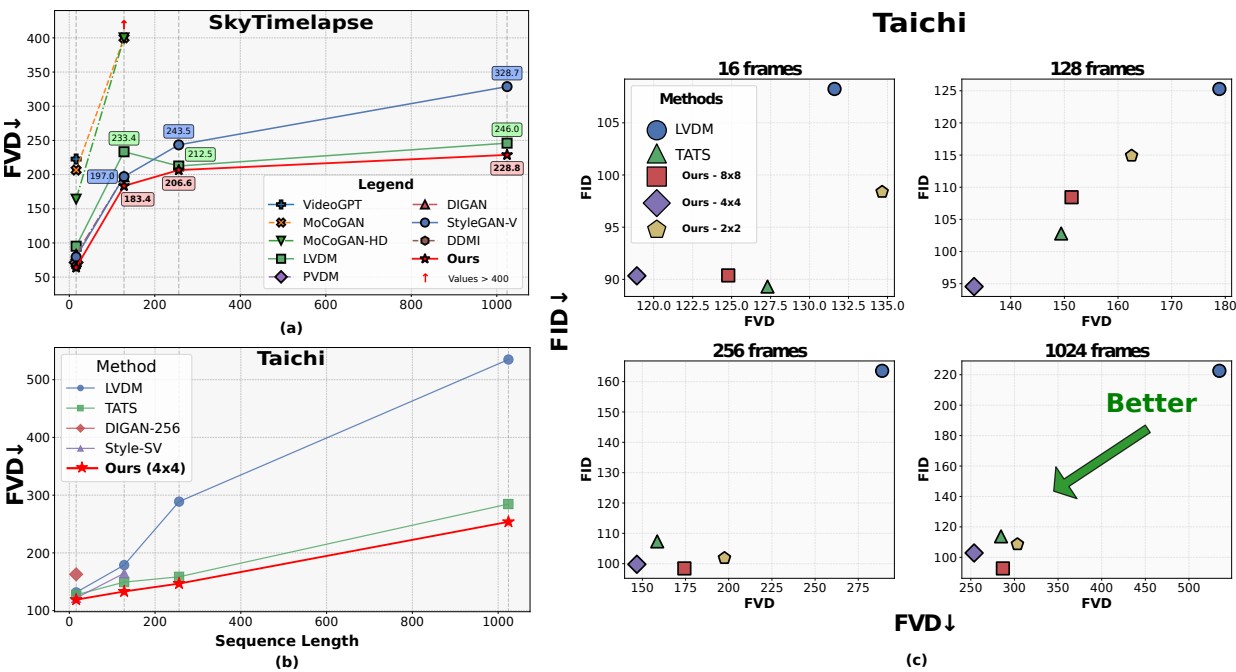

Figure 6: Quantitative comparisons with the SoTA for arbitrary length generation on the **(a)** SkyTimeLapse **(b)** Taichi datasets. We observe that our method outperforms the SoTA convincingly, and our advantage over the SoTA increases monotonically with increasing sequence length. **(c)** We study the effect of varying the grid size ($K$) on the observed FVD and FID on synthetic Taichi videos.

Table 3: We present a consolidated analysis of the **sampling time (s)** achieved by our method and its variants in different video length settings on the SkyTimelapse and Taichi datasets at $256 \times 256$ resolution. Our method massively outperforms the SoTA in all settings and the comparative advantage grows with increasing video length. We report the highlighted rows as our method's benchmark considering all associated tradeoffs. (**Bold**: best entry. underline: second best entry.)

| Method | Video Length (SkyTimelapse) | | | | Method | Video Length (Taichi) | | | |
|---|---|---|---|---|---|---|---|---|---|
| | 16 | 128 | 256 | 1024 | | 16 | 128 | 256 | 1024 |
| VideoGPT | 58.7 | - | - | - | LVDM | 91.7 | 273.4 | 284.8 | 1204.1 |
| MoCoGAN-HD | 77.8 | - | - | - | TATS | 95.8 | 287.5 | 482.6 | 1308.4 |
| LVDM | 63.7 | 112.8 | 245.4 | 1186.1 | Style-SV | 86.5 | 241.4 | - | - |
| PVDM | 47.6 | 168.0 | - | - | Ours - 2x2 | 42.3 | 206.8 | 307.2 | 1380.0 |
| StyleGAN-V | 62.0 | 243.2 | 386.3 | 1467.2 | **Ours - 4x4** | 4.8 | 38.27 | 79.66 | 515.43 |
| **Ours** | **4.8** | **38.27** | **79.66** | **515.43** | Ours - 8x8 | 4.8 | 15.0 | 26.5 | 51.4 |

they perform qualitative assessments typically only up to 128 frames. In contrast, in Figure 6 we compare our method with the SoTA on the (a) SkyTimelapse and (b) Taichi datasets for sequence generation up to 1024 frames at $256 \times 256$ resolution. We also present the data shown in Figure 6 (a) in tabular form in Appendix A.9 for the 16 and 128 length video settings for visual comfort. On CT-RATE, we compare with GenerateCT, a specialized 3D CT generator that utilizes a three-stage architecture, CT priors, and language guidance for generating 3D CT volumes (comprising 201 slices) at a resolution of $512 \times 512$ in Table 1. We utilized our Grid-based Autoregressive sampling to generate all the sequences used in our comparisons. The results demonstrate the following: **(1)** We conclusively outperform prior approaches comprising different modeling paradigms, viz. GANs, INRs, Diffusion models, and their combinations. **(2)** Our approach is

Table 4: We observe that our 3D positional embeddings yield superior perceptual quality (FVD, FID) and sptiotemporal consistency (FVD) than conventional 2D positional embeddings across all experimental setups and sequence lengths. (↓: lower is better. **Bold**: best entry.)

| Setup | Resolution | Pos Emb. | FVD-16 ↓ | FID-16 ↓ | FVD-128 ↓ | FID-128 ↓ | FVD-201 ↓ | FID-201 ↓ |
|---|---|---|---|---|---|---|---|---|
| CT-RATE | 512 | 2D | 289.74 | 58.8 | 378.2 | 56.7 | 1074.8 | 55.6 |
| CT-RATE | 512 | 3D | **268.3** | **53.3** | **356.1** | **55.2** | **998.9** | **54.8** |
| SkyTimelapse | 256 | 2D | 71.2 | 41.3 | 199.6 | 44.8 | - | - |
| SkyTimelapse | 256 | 3D | **64.1** | **37.7** | **183.4** | **40.9** | - | - |
| Taichi (4×4) | 256 | 2D | 139.8 | 159.5 | 152.6 | 167.6 | - | - |
| Taichi (4×4) | 256 | 3D | **118.9** | **157.3** | **133.2** | **164.6** | - | - |

domain-agnostic and does not require additional priors or supervision to model unconventional data. **(3)** Our design elements come together to support faithful arbitrary-length general image sequence synthesis. **(4)** Despite training only on sequences having $\leq 400$ frames, our method generates much longer high-fidelity sequences. Moreover, our relative superiority over other methods becomes more pronounced as the length of sampled sequences increases. Thereby confirming robust generalization, free of leakage (Somepalli et al., 2023) or memorization artifacts (van den Burg & Williams, 2021).

Figure 5 shows qualitative comparisons with the SoTA for standard video generation. For GenerateCT we used the prompt '44 years old male: The overall examination is within normal limits' to sample. All other sequences shown in the figure were sampled unconditionally. Our method exhibits improved sequence consistency and sharpness in CT volumes compared to the SoTA. GenerateCT's volumes show random jumps (e.g., frames $12^{th}$ to $13^{th}$ and $15^{th}$ to $16^{th}$). On SkyTimeLapse, LVDM produces blurry videos with low variability, while StyleGAN-V generates unrealistic lighting. Our approach avoids these issues, producing more coherent and realistic samples. Since we model motion at low resolution, we must investigate the potential artifacts that could emerge due to insufficient modeling resolution and potentially lossy super-resolution. To that end, we compare our method with Diffusion Forcing (Chen et al., 2025) on the Minecraft dataset (Yan et al., 2023). The reasons for our experimental choice are twofold. First, the Minecraft dataset comprises gameplay videos that contain large amounts of motion content per frame. Second, Diffusion Forcing belongs to a recent class of literature that intervenes with the Diffusion noise scheme in Autoregressive video generation. Consequently, comparing with it ensures completeness in our evaluation. Here, we remark that although Minecraft videos warrant a $128^2$ resolution only, we chose our $\{8 \times 8$ grid, four row control signal$\}$ setting for step 1 of sampling to intentionally allow room for $\times 2$ SR, ensuring fair comparison. We report these experimental results in Figure 5 and Table 2. Therein, we employ the average $l_1$ distance between consecutive frames as a metric for flicker, following Yang et al. (2024) who use it in the same context. We found that our method performs comparably to Diffusion Forcing at shorter synthetic sequence lengths and outperforms it at longer sequence lengths. The trend is consistent both in terms of synthesis quality and flicker. Thereby establishing that our method's limitation of modeling motion only at low resolution does not become a handicap even when generating sequences with large amounts of motion on the Minecraft dataset. In essence, our results on datasets with high variability, such as Taichi and Minecraft, underscore the efficacy of our method in sampling arbitrarily long synthetic sequences from most real-world datasets, provided their frame resolution, motion content, and degree of variability lie within the bounds of those quantities in our studied datasets.

**Synthesis Efficiency.** Our method is more efficient than previous methods in: (1) **inference speed**; As reported in Tables 3 and 1, our model is consistently $> 2\times$ faster than SoTA across all three data domains and across variable sequence dimensions. We do observe, however, that our advantage over the SoTA reduces with increasing sequence length due to iterative SR that is slower. (2) **training data required**; we investigate this property in Table 1 wherein we observe that our method attains superior FVD and comparable slice-wise FID scores when compared with GenerateCT on the CT-RATE dataset with as little as 10% of the training data. Moreover, the performance improves monotonically with increasing data. (3) **simplicity**; in the case of the CT-RATE dataset, our approach is significantly simpler in contrast to GenerateCT, which employs a three-stage CT-prior dependent approach, requiring text conditioning to achieve the reported quality metrics.

### 4.3 Ablation Studies

**Grid Size** $(K)$**.** In Figure 6 (c), we analyze the effect of varying $K \in \{2, 4, 8\}$ in training and step 1 of sampling on the attained synthesis quality for videos sampled via *one, three, and four* rows as control signals in step 1 and interpolating half the rows in step 2 of sampling via the Grid-based Autoregressive Sampling algorithm. We conducted this experiment using different image sequence lengths on the Taichi dataset. We observe that: (1) there exists a tradeoff between the amount of long-range sequence modeling signal and frame-wise fine information that a grid size setting has to offer. For instance, $K = 8$ offers a higher temporal span for DiT's self-attention mechanism but causes significant loss of finer details in the subsampled grid elements. Whereas $K = 2$ is limited in the time field, it does not cause any loss of high-frequency information, as $H = 512$, $W = 512$, and the required frame resolution is 256. Despite that, $K = 8$ offers superior FVD and FID over $K = 2$ for all sequence lengths. Thereby, establishing that superior sequential modeling is more instrumental than resolution preservation for sequence generation with our method. (2) Our method gains in sequence modeling while sacrificing a little on the quality of individual frames. The same is reflected in the FVD versus FID tradeoff we observe here. (3) The setting $K = 4$ sits at a sweet spot between both tradeoffs, consequently, making it our setting of choice for obtaining most results.

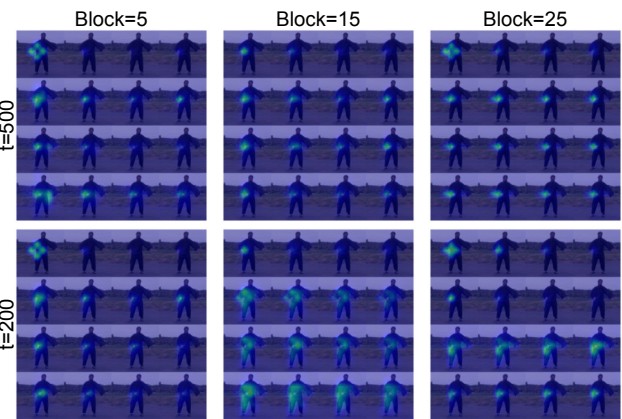

Figure 7: We visualize our Stage 1 (①) model's $1024 \times 1024$ attention maps scaled and overlayed on a corresponding sampled Taichi grid image. The emergence of high attention scores in grid-like patterns, bearing a direct correlation with the synthetic grid elements, suggests a strong self-attention prior that is key to our generation and sampling pipeline. ($t$ : reverse diffusion timestep. Block: $r$ denotes the $r^{th}$ DiT block.)

**Positional Embedding.** We observed certain 'looping artifacts' when generating sequences with our method using the vanilla DiT. In that, the frames would move back and forth in terms of motion, rather than having smooth transitions. We posit that the behavior was caused by the DiT's use of 2D positional embeddings, which are not suitable for consistently modeling motion across frames (subsampled grid elements in our case). To that end, we used 3D positional embeddings in our method as outlined in section 3.2. In Table 4, we justify our choice of 3D positional embeddings, demonstrating that they consistently benefit our method across all settings. We also observe the same result qualitatively, with complete remediation of the 'looping artifacts'. We present additional details in Appendix A.6.

**Role of refinement via SR.** We investigate the contribution of our SR (Stage-2 ②) model to the overall synthesis quality achieved by our method by contrasting the performance of our trained SR models with that of a SoTA off-the-shelf SR method, SinSR (Wang et al., 2024). We observe that our stage-2 outperforms SinSR in all settings, most importantly on the CT-RATE dataset, suggesting it is necessary to finetune the SR method in out-of-distribution settings (e.g., super-resolving medical images). We provide more details in Appendix A.8.

**Mechanistic Insights.** The DiT's strong self-attention mechanism is crucial to our approach since the generation of coherent grid images and the Grid-based Autoregressive Sampling algorithm both heavily rely on it. Consequently, we visualize attention maps from our Stage-1 (①) model, scaled and overlaid onto synthetic grid images in Figure 7 to gain a better understanding of the attention mechanism in our method's context. Therein, we observe that high attention scores emerge in grid-like patterns, suggesting that regions within grid elements attend mostly to corresponding regions within other grid elements. This thereby forms

the basis of the observed sequence-wise consistency and provides evidence for the soundness of our grid-based formulation.

We defer additional experimental results to the appendices. Specifically, we explore our model's ability to function as a plug-and-play image restoration model in Appendix A.12, where we perform 3D CT volume denoising using our method. Our diffusion-based approach outperforms previous baselines on this task. We also provide additional experimental evidence supporting the unique positioning of our approach within the image sequence generation landscape in Appendix A.11. Finally, we include our synthetic videos for further analysis in our supplementary material (Appendix B), as well as additional qualitative results in Appendix C.

## 5 Limitations and Future Work

By modeling sequences as grids of subsampled images, we incur certain losses in capturing the fine motion between frames. Even though these losses are insignificant on most real-world datasets, as shown in our experiments, they could potentially lead to inconsistencies for unobservable moving objects at low resolution. Along similar lines, the interpolation involved in our sampling scheme may cause undesired smoothening for video datasets captured at significantly higher frame rates than most real-world datasets. We posit that extending our formulation by adding explicit conditioning for the desired output frame rate shall make it adaptable to the demands of other datasets with finer motion as well. Thereby enhancing its overall utility. Additionally, our autoregressive long-form video generation relies on naive diffusion inpainting, which could be improved with more efficient algorithms. We have not been able to study the scaling laws for our models presented in this work due to computational constraints. It would be interesting to scale our models to the multi-billion parameter size group and compare their performance with relevant baselines currently excluded from our scope of analysis. Moreover, it is important to further validate our method's performance on datasets with higher frame resolution and motion content than those of our studied benchmarks. Addressing these limitations, incorporating multi-modal conditioning, and ethics and safety studies constitute a promising design space for future work. We further elaborate upon the societal implications of our work in our detailed impact statement in Appendix A.10.

## 6 Conclusion

Image sequences have historically been treated as large tensors of stacked frames. As a result, fixed-length synthesis, subpar sequential coherence, and prohibitively slow sampling rates have been some of the long-standing limitations in their generation. We consider them as grid images comprising subsampled frames instead, which are later super-resolved back to the original resolution. Therefore, being able to effectively harness modern self-attention based architectures and autoregressive sampling for the task whilst gaining on efficiency. Our method offers superior synthesis quality, efficiency, and support for generating arbitrary-length sequences without relying on proxy approximators, such as INRs. It generalizes well to specialized domains, such as 3D CT volumes, without prior-driven designs. The presented ablation studies show that GriDiT's underlying mechanisms conform to its theoretical formulation and substantiate our design choices. Overall, GriDiT establishes a strong framework for scalable image-sequence generation and opens avenues for improved pixel-space data representations in future research.

## 7 Acknowledgments

This work was funded by the National Institutes of Health (NIH) Grant No. R01EB032716 and NSF grants IIS-2123920, IIS-2212046.

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

# A   Appendix

## A.1   Related Work

**Diffusion Models.**  The superior generation quality of diffusion models has made them the de facto paradigm of choice for image and sequence generation (Rombach et al., 2022; Blattmann et al., 2023). Diffusion models, originated from score-based models (Hyvärinen, 2005; Song & Ermon, 2019) and made popular with Denoising Diffusion Probabilistic Models (DDPMs) (Ho et al., 2020), train a denoiser network that learns to reverse a corruption process that adds Gaussian noise to the data. The first diffusion pipelines used convolutional U-Nets (Ronneberger et al., 2015) as the denoiser architecture. We employ the **Diffusion Transformer (DiT)** (Peebles & Xie, 2023) architecture for our denoiser networks. DiT is the current SoTA on image generation. Instead of a U-Net denoiser, DiTs utilize a series of blocks with multi-headed self-attention. Another critical aspect of DiT is converting 2D images into a 1D sequence of tokens by patchifying the image, computing patch embeddings with positional encoding and ordering them in a sequence. This process allows the DiT to learn a strong self-attention prior over all spatial regions within the image. The prior forms the bedrock of our method as shown in later sections.

**Video Generation.**  Despite its importance, video generation lags behind image generation, primarily due to the high computational cost of processing large video tensors (He et al., 2022; Yan et al., 2021; Tulyakov et al., 2018; Tian et al., 2021; Yu et al., 2023; Skorokhodov et al., 2022; Yu et al., 2022; Park et al., 2024; Brooks et al., 2022; Girdhar et al., 2025; Guo et al., 2024). Most methods (Yan et al., 2021; Tulyakov et al., 2018; Tian et al., 2021; Brooks et al., 2022; Blattmann et al., 2023; Girdhar et al., 2025) model videos as large tensors, limiting maximum sequence length and incurring slow inference rates. Recent works using DiTs (Cai et al., 2024; Bian et al., 2025; Dalal et al., 2025) focus on architectural improvements for better conditioning; we consider these concurrent, but orthogonal to our goal of rethinking image sequence modeling. Approaches leveraging proxy models such as INRs (Skorokhodov et al., 2022; Yu et al., 2022; Park et al., 2024; Gupta et al., 2024) trade off efficiency for perceptual quality, yet still face limitations in scalability. Factorized generation has shown promise: Emu Video and AnimateDiff (Girdhar et al., 2025; Guo et al., 2024) split text-to-video into text-to-image and image-to-video stages, while LongVideoGAN (Brooks et al., 2022) factorizes within a GAN-based framework but lacks support for arbitrary-length sequences or resolutions beyond $256{\times}256$.

There exists a paucity of methods (Skorokhodov et al., 2022; He et al., 2022; Yu et al., 2023; Ge et al., 2022) that attempt arbitrary length video generation. Of these, PVDM (Yu et al., 2023) and LVDM (He et al., 2022) are latent diffusion based approaches. Whereas TATS (Ge et al., 2022) uses a GAN and StyleGAN-V is a GAN approach paired with INRs. We compare with all these methods on the standard video generation task and with StyleGAN-V (Skorokhodov et al., 2022), LVDM (He et al., 2022), and TATS (Ge et al., 2022) on the arbitrary length video generation task to ensure that we cover the several different approaches taken to solve the problem. Ours is the first method to employ factorization in a self-attention powered diffusion regime for arbitrarily long image sequence generation to the best of our knowledge.

**Applications of multiscale methods in Generative Modeling.**  Multiscale methods have so far been employed in the literature on Generative Models for two primary reasons. First, they help make the involved computations tractable. Second, factorizing the ground truth probability distribution as a product of several independent probability distributions at each progressive scale helps improve the synthesis quality at the resultant scale (resolution).

As far as images are concerned, the applications of multiscale methods in Generative Models go way back to the era of ProGAN and StyleGAN series of works (Karras et al., 2018; 2021b; 2020; 2021a; Tomar & Rajagopalan, 2022; Tomar et al., 2024; Tomar & Rajagopalan, 2023) wherein every progressive Generator layer added additional structure and style information from a coarse to fine scale to achieve high-quality outputs at high-resolution. However, these methods suffered from mode collapse despite their highly photorealistic synthesis quality. To that end, (Zhang et al., 2021) proposed a generative network that leverages a multiscale structure to solve high-dimensional Bayesian inference, thereby better addressing mode collapse in GANs. Growing interest in score-based generative modeling has prompted an impetus to accelerate the time-reversed

discrete Stochastic Differential Equations (SDEs) employed by Diffusion Models. Consequently, (Guth et al., 2022) proposed a method that generated increasingly higher-resolution images by discretizing reverse diffusions on wavelet coefficients at each scale. In a similar spirit, (Zhang et al., 2021) demonstrated that score estimation for large, complex images can be reduced to low-dimensional Markov conditional models across scales, thereby making the computations tractable. Whereas (Guth et al., 2023) proposed factorizing the data distribution into a product of conditional probability distributions that are strongly log-concave, this approach addresses mode collapse in generative models to some extent. (Mei, 2025) took a particularly distinctive take on the subject, wherein they showed that U-Nets (Ronneberger et al., 2015) can naturally implement the belief propagation denoising algorithm in generative hierarchical models (Li et al., 2000; Willsky et al., 2002; Jin & Geman, 2006).

Multiscale methods have also shown significant promise in the context of video generation. A plethora of works (Brooks et al., 2022; Ge et al., 2022; Yu et al., 2022; Villegas et al., 2023; Yan et al., 2024) have established their efficacy in making computations tractable while also improving synthesis quality. Of these, (Brooks et al., 2022) is a direct approach that splits generation across multiple resolutions. At the same time, others are indirect and work by factorizing the involved probability distributions in different ways. A series of closely related works Ho et al. (2022); Singer et al. (2023); Blattmann et al. (2023) have also attempted to extend an image generator to an image-sequence generator. However, they either rely on architectural changes and additional modules to accomplish the task or, use the image generator to only get meaningful priors for an entirely different image-sequence generator.

Our use of a Grid-based representation to implement a multiscale factorization of the probability distributions involved in image-sequence generation sets us apart from the prior art in the domain. Moreover, none of these previously proposed multiscale video generation approaches can directly leverage a *vanilla image generator* to generate *image-sequences.*

**Autoregressive sampling.** Autoregressive (AR) sampling entails employing information from previously generated samples to generate new samples. The rise of self-attention powered transformers (Vaswani et al., 2017) has led to a wide array of AR generation applications (Vaswani et al., 2017; Brown et al., 2020; Yang et al., 2019) in Natural Language Processing (NLP). However, the technique remains under-utilized in the image sequence generation context, with only a handful of methods (He et al., 2022; Yan et al., 2024; Deng et al., 2025; Yan et al., 2021) making use of it. Of these, LVDM (He et al., 2022) is the most closely related to our work as it attempts to employ AR sampling for arbitrarily long video generation. Although, it does not make use of a grid-based formulation. We recognize NOVA (Deng et al., 2025) as a concurrent work and MAGVIT-v2 (Yan et al., 2024) employs Large Language Models (LLMs) for video generation which is an orthogonal research direction. Therefore, we omit NOVA (Deng et al., 2025) and MAGVIT-v2 (Yan et al., 2024) in our comparative studies.

**Image Sequence Generation.** We address image sequence generation beyond conventional video tasks. For instance, lung CT (Falta et al., 2023; Gao et al., 2024; Song et al., 2024; Jeong et al., 2023) and brain MRI (Young et al., 2024) in 3D medical imaging, where modeling inter-slice dependencies is critical. Regarding 3D CT volume synthesis, GenerateCT (Hamamci et al., 2024c) is the only method reporting spatiotemporal consistency on public 3D CT data, making it our primary baseline. Notably, GenerateCT employs a complex, text-conditional three-stage pipeline. Whereas our approach is simple and unconditional.

### A.1.1 Concurrent work and Very Large-scale Models.

We treat very large-scale models ($\geq$ 2B parameters) such as Open-Sora (Zheng et al., 2024; Peng et al., 2025), Hunyan Video (Kong et al., 2024), CogVideoX (Yang et al., 2024), FIFO-diffusion (all variants implemented by the authors) (Kim et al., 2024), (Yang et al., 2024), and VAR (Tian et al., 2024) beyond the scope of comparison with this work. We also exclude comparisons with models trained on combinations of multiple datasets or those trained on a single very large (more than a few hundred thousand datapoints) datasets. We make this choice for two reasons. First, it is computationally intractable to work with them within our compute budget. Second, the comparison is unfair for our model. As far as concurrent work is concerned, LATTE (Ma et al., 2025) stands out among the plethora of related works. Although it is similar to our

approach in using the DiT (Peebles & Xie, 2023), it's objective of quality maximization of fixed-length videos is fundamentally different to ours. Our approach is more about devising a data modeling scheme that supports arbitrary-length, efficient, and generalizable image-sequence generation than about fixed-length video quality maximization.

## A.2 Preliminary on Denoising Diffusion Models

We employ Denoising Diffusion Probabilistic Models (DDPMs) (Ho et al., 2020) to learn and sample from the target distributions. DDPMs generate samples by learning to invert the process of information corruption by adding Gaussian noise. The forward diffusion corrupts the data, which the learned model reverses to synthesize new samples. The forward process is characterized by: $x_t = \sqrt{\bar{\alpha}_t}x_0 + \sqrt{1 - \bar{\alpha}_t}\epsilon_t$, where $\epsilon_t \sim \mathcal{N}(0, \mathbf{I})$ and the reverse process is characterized by: $\mathbf{x}_{t-1} = \frac{1}{\sqrt{\alpha_t}}\left(\mathbf{x}_t - \frac{1-\alpha_t}{\sqrt{1-\bar{\alpha}_t}}\epsilon_\theta\left(\mathbf{x}_t, t\right)\right) + \sigma_t\mathbf{z}$, where $z \sim \mathcal{N}(0, \mathbf{I})$ and $\epsilon_\theta$ is the noise predicted by the learned model parametrized by $\theta$.

## A.3 Model Architectures

Our Stage 1 (①) model is a DiT (Peebles & Xie, 2023) wherein class conditioning is removed, and only timestep conditioning is retained when used as a denoiser in our DDPM (Ho et al., 2020) training process. The hyperparameters employed in training Stage 1 are listed below:

- DiT variant: DiT-XL/2

- Training resolution: 512

- Model depth: 28

- Embedding dimension: 1152

- Patch size: 2

- Number of self-attention heads: 16

Whereas Stage 2 (②) is a vanilla DiT with appropriate modifications to use low-res (degraded) images as class conditioning. The subtle modifications are outlined below. Stage 2 shares the same hyperparameters as stage 1.

### A.3.1 Specifics of our Stage 2 architecture for Super-resolution

As illustrated in Figure 2 (b), we super-resolve the low-resolution frames using a conditional DiT model. In this case, we use the DiT model with certain modifications since we are generating a single, high-resolution (HR) image, conditioned on its low-resolution (LR) counterpart. More specifically, we obtain LR images from our training datasets of HR images by performing a combination of degradations (noise addition and blurring) on the HR images. Our training dataset for the task now comprises several {LR, HR} pairs. For each pair, the SR model's goal is to learn to generate HR images conditioned on the corresponding LR input. We train it to do so by embedding both LR and HR images in the VAE's (Kingma & Welling, 2022) latent space, concatenating the two latents, projecting the concatenated latent on the original hidden dimension, and training the DiT to generate the embedding for HR given the obtained projected embedding as input. We apply conditioning to the DiT via adaptive layer norm. We summarize the process of going from a coarse synthetic grid image comprising subsampled frames to highly photorealistic and motion-preserving individual frames in Figure 3.

## A.4 Grid-based Frame Modeling - a formal perspective

Here, we elaborate upon the process of obtaining grid images from image sequence data. These grid images are later used in the low-res generation aspects of our work. As illustrated in Figure 2, given an $\bar{N} \times H \times W$ tensor data point, we first down-sample it to $\bar{N} \times H/K \times W/K$ dimensions using bicubic interpolation. Subsequently,

we extract $K^2$-length sub-sequences along the tensor's channel dimensions such that the $n^{th}$ sub-sequence comprises indices $\{n, n+1, n+2 \ldots n+K^2\}$ along the channel dimension. Finally, for grayscale images, all elements of the sub-sequence are repeated three times along the channel dimension and concatenated, while

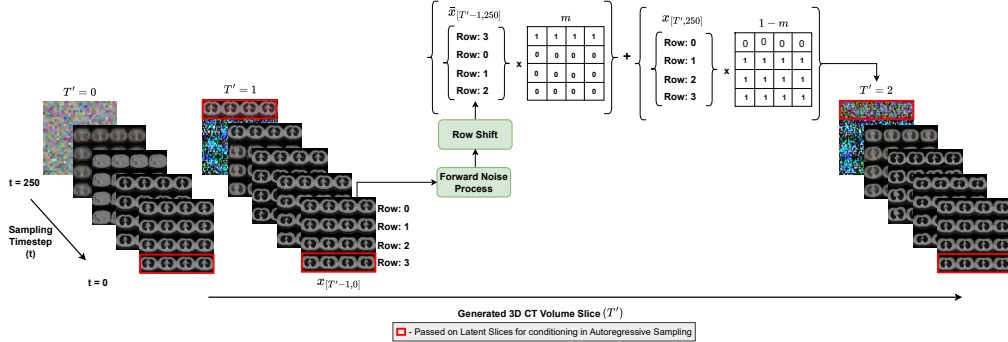

Figure 8: We illustrate the `Row Shift and Masking` operations used to form the *control signal* in our Grid-based Autoregressive sampling algorithm to transfer the last row from the previous iteration as the first in the next one while sampling grid images corresponding to subsampled synthetic 3D CT Volumes similar to ones from the CT-RATE dataset. The one-row control signal is only shown for brevity and clarity. In practice, we use three-row control signals in step 1 sampling for all our experiments.

preserving their ordering, to form a grid tensor with dimensions $3 \times H \times W$. Therein, the grid elements $(0,0)$, $(0,1)$, and $(K-1, K-1)$ denote the CT slices indexed by $n$, $n+1$, $n+K^2$ along the channel dimension of the sub-sequence, respectively.

## A.5 Additional details on Grid-based Autoregressive sampling

---

**Algorithm 1** Grid-based Autoregressive Sampling Step 1 (Coarse Generation)

---

**Input:** $\mathbf{X}_{[0,0]}$ ▷ The first grid image generated via standard DDPM sampling for $\mathcal{T}_s$ steps.
**Output:** $\mathbf{V}'$ ▷ A coarsely-coherent sequence of grid images starting with $\mathbf{X}_{[0,0]}$.

1: $\mathbf{V}' \leftarrow \{\}$
2: **for** $T' = 1, 2, \ldots, N$ **do**
3:      $\mathbf{X}_{[T',\mathcal{T}]} \sim \mathcal{N}(0, I)$
4:      **for** $t = \mathcal{T}_s, \mathcal{T}_s - 1, \mathcal{T}_s - 2, \ldots, 1$ **do**
5:          **if** $t > 1$ **then**
6:              $\epsilon \sim \mathcal{N}(0, \boldsymbol{I})$
7:          **else**
8:              $\epsilon \leftarrow 0$
9:          **end if**
10:          $\bar{\mathbf{X}}_{[T'-1,t-1]} \leftarrow \sqrt{\bar{\alpha}_t}\, \mathbf{X}_{[T'-1,0]} + \sqrt{1 - \bar{\alpha}_t}\, \epsilon$
11:          $\bar{\mathbf{X}}_{[T'-1,t-1]} \leftarrow \mathrm{RowShift}(\bar{\mathbf{X}}_{[T'-1,t-1]})$
12:          $\mathbf{X}_{[T',t-1]} \leftarrow \sigma_t\, \epsilon + \frac{1}{\sqrt{\alpha_t}}\left( \mathbf{X}_{[T',t]} - \frac{\beta_t}{\sqrt{1 - \bar{\alpha}_t}}\, \epsilon_{\theta_1^*}(\mathbf{X}_{[T',t]}, t) \right)$
13:          $\mathbf{X}_{[T',t-1]} \leftarrow (m \odot \bar{\mathbf{X}}_{[T'-1,t-1]}) + ((1-m) \odot \mathbf{X}_{[T',t-1]})$
14:      **end for**
15:      $\mathbf{V}' \leftarrow \mathbf{V}' \cup \{\mathbf{X}_{[T',0]}\}$
16: **end for**
17: **return** $\mathbf{V}'$ ▷ Serves as input to Step 2.

---

**Notation pertaining to our sampling scheme.** $\mathbf{X}_{[T',t]}, \mathbf{X}_{[T',t]}, \mathbf{X}_{prev}, \mathbf{X}_{next}, \bar{X}, m, m_{prev}, m_{current},$ and $m_{next}$ denote a sample at iteration $T'$ and diffusion timestep $t$ in step 1, a sample at iteration $T''$ and diffusion

timestep $t$ in step 2, the previous sample from step 1 at iteration $T''$ of step 2, the next sample from step 1 at iteration $T''$ of step 2, forward noised version of a sample $X$, a binary mask representing a grid with

---

**Algorithm 2** Grid-based Autoregressive Sampling Step 2 (Interpolation for Temporal Super-resolution)

---

**Input:** $\mathbf{V}' = \{\mathbf{X}_{[T',0]} : T' \in [0, N]\}$ ▷ The output sequence from Step 1.
**Output:** $\mathbf{V}''$ ▷ A spatially coarse sequence of grid images with superior temporal resolution than Step 1.

1: $\mathbf{V}'' \leftarrow \{\}$
2: **for** $T'' = 0, 1, \ldots, N-1$ **do**
3:      $\mathbf{X}_{[T'',\mathcal{T}]} \sim \mathcal{N}(0, I)$
4:      $T' = T''$
5:      $\mathbf{X}_{prev} = \mathbf{X}_{[T',0]}, \mathbf{X}_{next} = \mathbf{X}_{[T'+1,0]}$
6:      **for** $t = \mathcal{T}_s, \mathcal{T}_s - 1, \mathcal{T}_s - 2, \ldots, 1$ **do**
7:          **if** $t > 1$ **then**
8:              $\epsilon \sim \mathcal{N}(0, \boldsymbol{I})$
9:          **else**
10:             $\epsilon \leftarrow 0$
11:          **end if**
12:          $\bar{\mathbf{X}}_{prev} \leftarrow \sqrt{\bar{\alpha}_t}\,\mathbf{X}_{prev} + \sqrt{1 - \bar{\alpha}_t}\,\epsilon$
13:          $\bar{\mathbf{X}}_{next} \leftarrow \sqrt{\bar{\alpha}_t}\,\mathbf{X}_{next} + \sqrt{1 - \bar{\alpha}_t}\,\epsilon$
14:          $\mathbf{X}_{[T'',t-1]} \leftarrow \sigma_t \epsilon + \frac{1}{\sqrt{\alpha_t}}\left(\mathbf{X}_{[T'',t]} - \frac{\beta_t}{\sqrt{1-\bar{\alpha}_t}} \epsilon_{\theta_1^*}(\mathbf{X}_{[T'',t]}, t)\right)$
15:          $\mathbf{X}_{[T'',t-1]} \leftarrow (m_{prev} \odot \mathrm{RowShift}(\bar{\mathbf{X}}_{prev})) + (m_{current} \odot \mathbf{X}_{[T'',t-1]}) + (m_{next} \odot \bar{\mathbf{X}}_{next})$
16:      **end for**
17:      $\mathbf{V}'' \leftarrow \mathbf{V}'' \cup \{\mathbf{X}_{[T'',0]}\}$
18: **end for**
19: $\mathbf{V}'' \leftarrow \mathbf{V}' \cup \mathbf{V}''$ ▷ Frames obtained by splitting the grid images in $\mathbf{V}'$ and $\mathbf{V}''$ and retaining unique elements only are combined by inserting newly interpolated frames between previously generated frames.
20: **return** $\mathbf{V}''$

---

$K = 4$ having first three rows set to 1 and last one set to 0, a similar binary mask having first row set to 1 and rest set to 0, a similar binary mask having rows 1,4 set to 0 and rows 2,3 set to 1, a similar binary mask having last row set to 1 and rest set to 0.

**A Formal Perspective on Our Sampling Scheme.** As summarized by Algorithms 1 and 2, the process of generating the grid-image $T'$ in Step 1 starts with generating the first grid image $\mathbf{X}_{[0,0]}$ per the vanilla DiT sampling procedure. This is followed by several autoregressive sampling iterations until the grid image $T'$ is arrived at. Each reverse diffusion timestep $t$ of every autoregressive iteration $T'$ entails: (1) adding appropriate noise to the previous generated grid image $\mathbf{X}_{[T'-1,0]}$ per the forward process, (2) obtaining its row-shifted and masked version $\bar{\mathbf{X}}_{[T'-1,t-1]}$, (3) combining that version with a tensor obtained by denoising $\mathbf{X}_{[T',\mathcal{T}]} \sim \mathcal{N}(0, \boldsymbol{I})$ for $t-1$ timesteps. The process continues for $\forall t \in [0, \mathcal{T}]$ and $\forall T' \in [1, N]$. In effect, the binary mask $m$ acts as a gating mechanism that decides the amount of previously generated information used at a particular autoregressive sampling iteration. All operations are performed in the DiT latent space, with scale factors applied for reduced dimensions. The latent encoding and decoding steps are omitted for brevity.

The core operations performed in step 2 are similar to step 1 except for the fact that they now result in interpolated frames due to the nature of ordering in our grid-based formulation and the model's priors.

The binary masks $m, m_{prev}, m_{current},$ and $m_{next}$ are a $K \times K$ matrices scaled appropriately to latent space dimensions with all elements as described in the Notation pertaining to our sampling scheme. This is further elucidated by Figure 8 which also illustrates the `RowShift` operation. We set $\mathcal{T}_f, H, W,$ and $K$ to be 250, 512, 512, and 4 in all our experiments on the CT-RATE dataset (Hamamci et al., 2024a;b). Whereas for experiments on the SkyTimelapse (Zhang et al., 2020) and Taichi (Siarohin et al., 2019) dataset, we change $H$ and $W$ to 256.

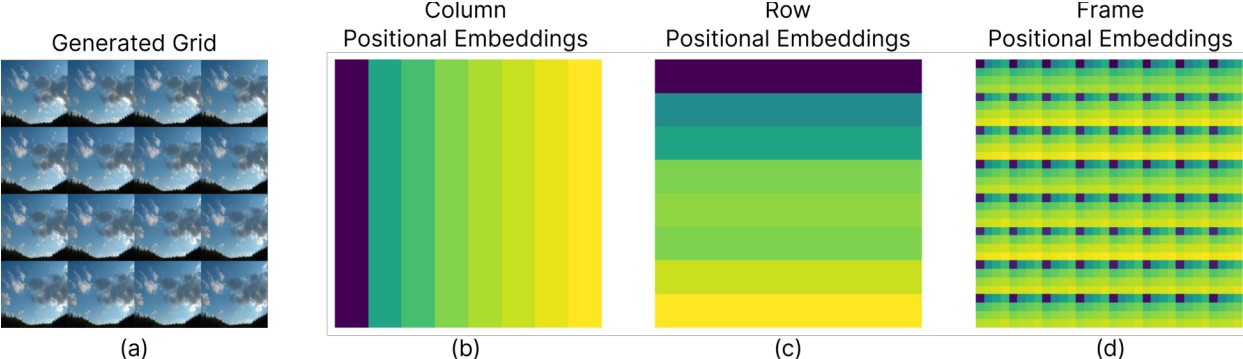

Figure 9: (a) In image grid generated from our SkyTimelapse model. In (b), (c), and (d), we contrast how the different positional embeddings encode information about the (x,y) location of each patch as well as its temporal location within the sequence. Here, (b) and (c) represent different 2D positional embeddings, whereas (d) represents 3D positional embeddings. We make use of the (d) 3D (frame) positional embeddings in our formulation to make sure the correspondence between grid elements is captured perfectly. Similarities between (a) and (d) are indicative of the suitability of 3D positional embeddings to capture sequential information structured as grids.

### A.6 3D positional embeddings

In Table 4, we rerun our experiments without the 3D positional embeddings that we propose adding to the image grids. Without encoding the position within the sequence, the generated patches exhibit worse temporal consistency, which is evident by the reduction in FVD observed. Further, in Figure 9 we visualize how the 3D positional embeddings we use encode both the spatial location and location within the sequence for each individual image patch the DiT operates on. Thereby playing a crucial part in our Grid-based formulation.

Formally, our 3D positional embeddings use a broad mathematical formulation similar to the one employed by fixed $sin - cos$ positional embeddings from Vaswani et al. (2017). Our embeddings differ only in that we embed each latent dimension separately, followed by stacking them together along the positional embedding dimension. The equations below summarize our embedding scheme:

$$\text{PosEmbed}(p_{d_l}, p_h, p_w) = \left[\mathbf{e}^{(d_l)}(p_d); \mathbf{e}^{(h)}(p_h); \mathbf{e}^{(w)}(p_w)\right] \in \mathbb{R}^D \tag{3}$$

$$\mathbf{e}(p) = \left[\sin(p \cdot \omega_0), \ldots, \sin\left(p \cdot \omega_{\frac{d}{2}-1}\right), \cos(p \cdot \omega_0), \ldots, \cos\left(p \cdot \omega_{\frac{d}{2}-1}\right)\right] \tag{4}$$

$$d_{d_l} = d_h = d_w = \left\lfloor \frac{D}{3} \right\rfloor \tag{5}$$

Wherein $\text{PosEmbed}(p_d, p_h, p_w)$ represents the overall positional embedding at a particular embedding dimension index $(d_l)$, latent height index $(h)$, and latent width index $(w)$ for a total embedding length $D$. Eq. 4 further elaborates upon each individual 1D embedding component in Eq. 3. Each individual-dimensional embedding bears a formulation similar to Vaswani et al. (2017) with $p$ being an indexing variable for positions. Whereas Eq. 5 clarifies that each embedding individual 1D embedding dimension is allotted an equal length.

### A.7 Experimental Setup

We carefully curate our experiments with a threefold objective: evaluate GriDiT's performance for high-quality, arbitrary-length image sequence synthesis across diverse data regimes; justify our design choices; and elucidate the mechanisms underlying our method. Here, we provide comprehensive information about the various settings and design choices employed during these experiments.

### A.7.1 Dataset Details

We conduct our evaluations on three significantly different datasets that are widely used in baselines established by the prior art. (1) **The SkyTimelapse dataset** (Zhang et al., 2020), which comprises timelapse videos of skies in different lighting scenarios and associated ground imagery. Performance on this dataset is primarily indicative of generating sequences with long-range temporal consistency and photorealism of frames, as the dataset does not contain rapid motion or occlusions. We work with this dataset at $256 \times 256$ resolution and use the provided train and test splits. (2) **The CT-RATE dataset** (Hamamci et al., 2024a;b) that contains several 3D CT Volumes collected from real patients. These Volumes are sequences wherein the axis of variation is non-temporal. Performing well on this dataset requires accurate modeling of high-frequency structure (anatomy) and textures, as well as long-range consistency. We learn to generate sequences from this dataset at a resolution of $512 \times 512$. We use exactly the same preprocessing, train split, and test split as GenerateCT (Hamamci et al., 2024c). (3) **The Taichi dataset** (Siarohin et al., 2019) that was originally proposed for human action recognition and requires accurate modeling of motion at both coarse and fine scales to produce high-quality results. We use the dataset in its standard $256 \times 256$ resolution using its train and test splits.

### A.7.2 Training

We train two DiT (DiT-XL/2) (Peebles & Xie, 2023) models corresponding to our Stage 1 (①) and Stage 2 (②) models, respectively, for each dataset as described in section 3.1. Both stages of all our models were finetuned starting from the DiT (Peebles & Xie, 2023) weights pretrained on ImageNet (Deng et al., 2009), with a linear warmup schedule taking the learning rate from $10^{-6}$ to $10^{-4}$ over $10^4$ warmup (Loshchilov & Hutter, 2017) iterations. Self-attention based model architectures have been shown (Shallue et al., 2018) to benefit from larger batch sizes. Therefore, we made use of engineering methods such as mixed-precision training (Micikevicius et al., 2018) and gradient accumulation (Andersson et al., 2022) in our mini-batch optimization to train at larger batch sizes than those permitted by hardware constraints. We borrowed other elements of the training recipe viz. the objective function and the optimizer from DiT (Peebles & Xie, 2023) to train our models. We used a single NVIDIA RTX A6000 GPU for training all models.

### A.7.3 Sampling and Evaluation

We use our Grid-based Autoregressive sampling algorithm paired with the SR method with different control signal settings for generating sequences in all our experiments. We evaluate our method on the quality of the synthesized sequences as well as their consistency. We chose the widely accepted FVD (Unterthiner et al., 2019) and FID (Heusel et al., 2017) metrics for our experiments. We use the I$_3$D (Carreira & Zisserman, 2017) feature extraction backbone for computing FVD. We employ the evaluation pipeline established by StyleGAN-V (Skorokhodov et al., 2022) for computing FVD and FID. We measure the inference time in seconds (s) on an NVIDIA A100 GPU as a metric for efficiency. We do not provide metrics for intractable methods, viz., methods that do not support the experiment in question, do not provide their implementation, or yield severely poor qualitative performance when implemented.

Finally, we bring forth two critical points that are important with respect to evaluation. First, we use unconditional Stage 1 sampling in all experiments in the paper. Second, in all experiments where the length of synthetic videos is greater than that of videos in the test set, we looped the test set videos cyclically to have them all reach the experimental length and computed metrics with respect to them.

### A.8 Custom Super-resolution

We present a quantitative comparison of our model's stage (②) with a SoTA diffusion-based SR method, SinSR (Wang et al., 2024), in Table 6. We finetuned SinSR for comparison on the CT-RATE dataset since CT Volume slices are reasonably out of distribution with respect to its training data. We used its pre-trained variant for other experiments. Our methods' superiority over the SoTA underscores its utility in making our method work. Furthermore, upon qualitative observation, we found that it is more suitable than the SoTA in retaining high-frequency details while performing SR. This attribute is hugely significant in critical domains

Table 5: Our model outperforms the SoTA on video generation on 16 and 128 length sequences from the SkyTimelapse dataset. Our model is significantly better in terms of perceptual quality and spatiotemporal consistency (FVD) and > 2.5× faster. (↓ : lower is better. † and * : numbers reported by StyleGAN-V Skorokhodov et al. (2022) and DDMI Park et al. (2024), respectively. **Bold**: best entry. underline : second best entry.)

| Method | 16 Frames | | 128 Frames | |
|---|---|---|---|---|
| | FVD↓ | Sampling Time (s) ↓ | FVD↓ | Sampling Time (s) ↓ |
| VideoGPT | 222.7[†] | 58.56 | - | - |
| MoCoGAN | 206.6[†] | - | 575.9[†] | - |
| MoCoGAN-HD | 164.1[†] | 77.8 | 878.1[†] | - |
| LVDM | 95.2[*] | 91.75 | 233.4 | 273.4 |
| PVDM | 71.46[*] | 47.6 | - | - |
| DIGAN | 83.11[†] | - | 196.7[†] | - |
| StyleGAN-V | 79.52[†] | 62 | 197[†] | 243.25 |
| DDMI | 66.25[*] | - | - | - |
| **Ours** | **64.078** | **4.8** | **183.4** | **92.55** |

Table 6: We contrast our SR module's performance with that of the SoTA (SinSR) at different SR scales. Our model outperforms the SoTA in all settings, signifying its efficacy. (↑: higher is better. ↓: lower is better. † : fine-tuned model metrics. * : vanilla model metrics. **Bold**: best entry.

| Method | SkyTimelapse (×2) | | CT-RATE (×4) | | Taichi (×2) | | Taichi (×4) | |
|---|---|---|---|---|---|---|---|---|
| | PSNR (dB)↑ | FVD-16↓ | PSNR (dB)↑ | FVD-16↓ | PSNR (dB)↑ | FVD-16↓ | PSNR (dB)↑ | FVD-16↓ |
| SinSR | 29.8[*] | 71.7[*] | 21.42[†] | 593.23[†] | 34.52[*] | 138.67[*] | 31.93[*] | 119.48[*] |
| **Ours** | **31.982** | **64.08** | **29.48** | **383.45** | **35.28** | **134.654** | **33.48** | **118.919** |

such as 3D CT Volume imaging. Thereby, establishing that generative SR furthers our proposed method's applicability to general image sequence synthesis.

### A.8.1 On the degradation scheme employed to train our Stage 2 model (②)

As described in section 3.4 of the paper, our stage 2 model learns to refine individual frames by performing the surrogate task of restoring appropriately degraded frames. To that end, ground truth frames are first downsampled and then upsampled by the required scaling factor to simulate degradation caused by lossy super-resolution. We also add variable amounts of Gaussian noise to each image in the training dataset to account for losses caused by learning to model motion at low resolution. Consequently, these 'degraded' frames are used as conditioning signals to generate their corresponding 'restored' frames or the original ground truth frames. We use bicubic interpolation in all our scaling operations. At inference, refining a low-resolution frame entails upsampling it via bicubic interpolation and then using it to condition the stage 2 model's generation that yields the corresponding high-resolution frame. Figure 11 illustrates the aforementioned approach. We outline the degradations performed on ground-truth frames to obtain corresponding noisy frames used as surrogates for upsampled low-res frames in the Python function `process_image_resize_noise_blur` presented in Listing 1. The code elucidates the procedure for performing the degradations necessary for learning to super-resolve frames at $128 \times 128$ resolution to $256 \times 256$ resolution.

### A.8.2 Qualitative Comparisons with SoTA off-the-shelf SR models

We provide qualitative comparisons with SinSR (Wang et al., 2024), a SoTA off-the-shelf SR model on the CT-RATE (Hamamci et al., 2024a;b) (see Figure 12) and the SkyTimelapse (Zhang et al., 2020) (see Figure 13) datasets to substantiate our findings. The figures demonstrate the superiority of our method and justify the need for custom generative SR.

Table 7: Additional Comparison with naive baselines to validate our unique positioning in the design space. **Bold:** best.

| Method | Dataset | FVD-16 ↓ | FVD-128 ↓ | FVD-256 ↓ |
|---|---|---|---|---|
| Naive Baseline I (Channel-wise Stacking) | SkyTimelapse (256x256) | 85.5 | 313.2 | 452.5 |
| **Ours** | | **64.1** | **183.4** | **206.6** |
| Naive Baseline II (3D VAE) | Taichi (256x256) | 129.32 | - | - |
| Ours | | **118.919** | - | - |

## A.9 Quantitative results for image sequence generation on SkyTimelapse

We provide quantitative metrics for 16 and 128 length video generation on the SkyTimelapse dataset (Zhang et al., 2020) in Table 5. This is in supplement to Figure 6 (a), wherein the metrics for certain poorly performing methods might be difficult to elicit from the given plot due to a fine scale on the y-axis. As is evident, our method comprehensively outperforms the SoTA on the task.

## A.10 Statement of Broader Impact

Our work learns to model image sequences in a generative setting. Therefore, it does entail the risk of being misused like any other photorealistic image or video generative model. Therefore, its authentic distribution and ethical usage are essential. We shall release our model through GitHub or Huggingface. Both of which follow best practices to maintain community standards for ethical usage. We shall also include a widely accepted license in our release to prevent irresponsible usage. We would like to remind the reader that we only claim that our synthetic 3D CT Volumes bear statistical and visual resemblance to the Volumes present in the CT-RATE dataset curated by Hamamci et al. (2024a). Consequently, their real-world medical utility is yet to be established. As a result, users should refrain from using our work for real-world healthcare applications unless approved by appropriate medical authorities. We defer evaluation of our results from a medical standpoint to future follow-up work. The fact that our method uses datasets wherein it is hard for any bias to creep in comforts us in the quality of our work. Moreover, our work could advance image sequence generation in unconventional fields as well, leading to newfound applications in different domains of science and society.

## A.11 Validating our approach's unique positioning

Pixel-space data representation reformulation has been a widely overlooked aspect in image-sequence generation design space, almost as if it were *hidden in plain sight*. In that context, GriDiT is uniquely positioned by being the first and only approach to capitalize on this aspect to address the synthesis quality versus efficiency tradeoff. We devote this section to taking a proof-by-contradiction approach to further validate our thesis. Specifically, we construct two naive baselines using the DiT using conventional data representations and contrast our performance with them. We present our experimental results in this regard in Table 7.

**Naive Baseline I: Channel-wise stacking paired with DiT.** We establish this baseline by simply using a vanilla representation comprising channel-wise stacked frames for each training datapoint. In essence, we treat them as tensors bearing shape $f \times \frac{H}{K} \times \frac{W}{K}$ wherein $f$ represents the number of frames. We use the DiT's 2D VAE to embed each frame into the latent space sequentially and stack those latents channel-wise. Subsequently, we modify the DiT's projection layers to work with the inflated channel dimensions and train the model with the same recipe as our model. At inference, we use an Autoregressive sampling algorithm similar to ours in every aspect except for using the last $\frac{f}{K}$ channels as the sampling control signal. Finally, we use the same stage-2 (②) as ours to ensure fairness of comparison.

**Naive Baseline II: 3D VAE paired with DiT.** In this case, we replace the DiT's 2D VAE with the 3D VAE used by CogVideoX (Yang et al., 2024). The modification allows us to directly embed a sequence tensor bearing shape $f \times \frac{H}{K} \times \frac{W}{K}$ to a single latent that can be used to train the DiT model. As with the other experiment, we pair this modified model with our stage-2 (②) model for performing SR. Since this baseline cannot provide a structured sampling control signal, we restrict our experimentation in this setup to

---

**Algorithm 3** Diffusion-driven 3D CT Volume Denoising

---

**Input**: $\mathbf{X}_{[n,0]}$ (Nosiy Image Latents)

**Output**: $\hat{\mathbf{X}}_0$ (The Denoised Latents)

1: **for** $t = \mathcal{T}, \ldots, 1$ **do**
2:      $\epsilon \sim \mathcal{N}(0, I)$ if $t > 1$, else $\epsilon = 0$
3:      $\mathbf{X}_{[n,t]} = \sqrt{\bar{\alpha}_{t+1}}\mathbf{X}_{[n,0]} + \sqrt{(1 - \bar{\alpha}_{t+1})}\epsilon$
4:      $\hat{\mathbf{X}}_{t-1} = \frac{1}{\sqrt{\alpha_t}}\left(\mathbf{X}_{[n,t]} - \frac{\beta_t}{\sqrt{1-\bar{\alpha}_t}}\epsilon_{\theta_1^*}(\mathbf{X}_{[n,t]}, t)\right) + \sigma_t\epsilon$
5: **end for**
6: **return** $\hat{\mathbf{X}}_0$

---

Table 8: Denoising 3D CT Volumes. We compare different denoising methods under varying noise levels. Our method outperforms the baselines in all settings, demonstrating superior denoising capability. ($\mathcal{N}(0, \sigma^2)$: Noise (degradation) process with mean $= 0$ and variance $= \sigma^2$, ↑: higher is better.)

| Denoising Method | $\mathcal{N}(0, 25)$ | | $\mathcal{N}(0, 100)$ | |
|---|---|---|---|---|
| | PSNR (dB)↑ | SSIM↑ | PSNR (dB)↑ | SSIM↑ |
| Bilateral Filtering | 16.5 | 0.308 | 16.12 | 0.294 |
| GenerateCT | 23.81 | 0.357 | 23.41 | 0.350 |
| **Ours** | **41.26** | **0.855** | **34.65** | **0.758** |

16-length sequences only. We made sure to only use chose to use the Taichi dataset for these experiments because the 3D VAE yielded highly accurate reconstruction for those videos quantified by a reconstruction PSNR of 27.58 dB.

Given the experiments in Table 7, we make the three key inferences. First, our approach outperforms these baselines convincingly, thereby establishing the non-trivial nature of our contributions. Second, the benefits of our approach do not stem entirely from the inductive biases captured by the DiT-based diffusion paradigm; rather, all our design elements come together nicely to achieve our performance metrics. Third, the grid-based formulation lends itself better to diffusion-inpainting inspired autoregressive generation than a vanilla channel-wise stacked representation, making it crucial for utilizing the self-attention prior learned by the DiT in our method. In essence, these experiments underscore the need for looking beyond conventional modeling approaches and focusing on devising better data representations in the domain.

## A.12 3D CT Volume Denoising

The fact that our method learns a strong self-attention prior to the denoising diffusion process and can work with arbitrary-length sequences prompted

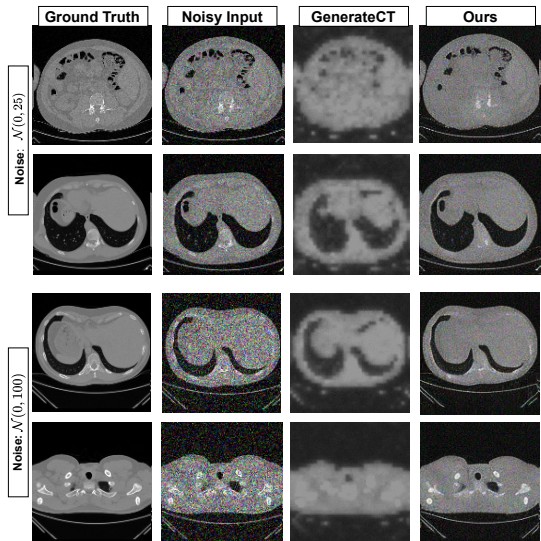

Figure 10: Employing our denoising diffusion pipeline for denoising 3D CT Volumes yields substantially superior performance than the baseline (Hamamci et al., 2024c). Thereby, suggesting a significantly stronger modeling ability, which enables our method to perform a task that is yet to emerge in the literature.

us to attempt image sequence (3D CT) Volume denoising. 3D CT volume denoising is a particularly relevant problem for two reasons. First, 3D CT Volumes are frequently corrupted by noise that creeps in due to

improper calibration of CT scan machines or improper preprocessing as mandated by the instrument type. Second, to the best of our knowledge, this problem has not been addressed by any other denoising diffusion-based method. Given a noisy CT volume, we start by converting a series of subsequences of the noisy volume to grid-images encoded into our latent space employed for diffusion, per our grid-based modeling procedure. We denote these latents as $x_{[n,0]}$. Subsequently, we denoise the latents corresponding to the noisy input signal per the procedure summarized by Algorithm 3. Effectively, we apply appropriate forward diffusion steps to the noisy signal and then perform reverse diffusion steps via our learned stage 1 model. Finally, we perform SR on the split grid elements using our learned Stage 2 and collate the slices together to get the denoised outputs. We prepare baselines for comparison using GenerateCT and standard bilateral filtering. GenerateCT does not support denoising. So, we used its diffusion-based Super-resolution block to perform denoising without text prompts to form a baseline. Our method outperforms GenerateCT and standard bilateral filtering in terms of denoised volume quality as presented in Table 8 and Figure 10, although it does lose out on a few high-frequency details in the ground truth image. These results are important as they further attest to our modeling and learning paradigm's efficacy in effectively representing image sequences.

Listing 1: Python code for performing the degradations required to train our stage-2 (SR) model.

```python
import os
import random
import cv2
import numpy as np

def add_gaussian_noise(image, mean=0, std=10):
    """Add Gaussian noise to an image."""
    noise = np.random.normal(mean, std, image.shape)
    noise = noise.astype(np.float32)
    noisy = cv2.add(image.astype(np.float32), noise)
    return np.clip(noisy, 0, 255).astype(np.uint8)

def process_image_resize_noise_blur(
    image_path,
    erosion_iterations=3,
    blur_radius=15,
    brightest_fraction=0.4,
    global_blur_radius=7):
    """
    Load an image, resize, add noise, and apply Gaussian blur
    with random parameters.
    """
    # Load and convert image to RGB
    image = cv2.imread(image_path)
    image = cv2.cvtColor(image, cv2.COLOR_BGR2RGB)
    # Resize image twice: 256x256 via 128x128
    image_resized = cv2.resize(
        cv2.resize(image, (128, 128), interpolation=cv2.INTER_CUBIC),
        (256, 256), interpolation=cv2.INTER_CUBIC
    )
    # Add Gaussian noise with random std
    std = random.randint(10, 15)
    noisy_image = add_gaussian_noise(image_resized, std=std)
    # Randomly apply Gaussian blur
    if random.random() < 0.5:
        result_image = noisy_image
    else:
        blur_sizes = [9, 11, 13, 15]
```

```
        blur_radius = random.choice(blur_sizes)
        result_image = cv2.GaussianBlur(
            noisy_image, (blur_radius, blur_radius), 0
        )

    return result_image
```

## B  Supplementary Material

We organize the contents of the supplementary material directory (submitted as a .zip archive) as follows:

- **Synthetic videos and their comparisons with the SoTA.** We provide several synthetic videos and their qualitative comparisons with the corresponding SoTA on the SkyTimelapse (Zhang et al., 2020) and Minecraft (Yan et al., 2023) datasets along with our synthetic 3D CT volumes in 'index.html'. We request the reader to please use their browser to view the same. the reader may please view the corresponding videos in their media player software if viewing in the browser is not feasible. The videos are stored in subdirectories organized as follows:
  - SkyTimelapse:                     'GriDiT_TMLR_supplementary_material/sky_videos', 'GriDiT_TMLR_supplementary_material/sky_long_video'
  - Minecraft: Our results are present in the file 'GriDiT_TMLR_supplementary_material/sky_videos /ours.mp4' and those obtained directly from the results released by DiffusionForcing (Chen et al., 2025) are present in the file 'GriDiT_TMLR_supplementary_material/sky_videos/diff_dorcing.mp4'.
  - Synthetic CT Volumes: 'GriDiT_TMLR_supplementary_material/ct_seq'
  - Please note that all elements arranged in a grid-form are individual videos independent of each other, and not the grid-elements as described in our formulation.

## C  Additional qualitative results

We present and analyze additional qualitative results to bring forth a better understanding of the pertinent aspects of our method.

### C.1  Arbitrary length video synthesis

We tie this discussion to the results presented in section 4.2 of the paper. The Taichi dataset (Siarohin et al., 2019) is a particularly challenging dataset from a video generation standpoint because it requires a model to infer large motion and high-frequency details from very few data points at a relatively low resolution ($256 \times 256$). Consequently, most prior methods in the domain struggle to get both motion and high-frequency details right on this dataset. In Figure 14, we compare our method's performance with the SoTA on the Taichi dataset (Siarohin et al., 2019), qualitatively. Therein, we make three key inferences. First, our method yields superior perceptual quality than both LVDM (He et al., 2022) and TATS (Ge et al., 2022) on arbitrarily long generation. We attribute this to the efficacy of our Grid-based Autoregressive sampling algorithm. We also notice a severe decline in quality with increasing sequence length in our competing methods. Such a decline indicates the overall worse modeling capability of approaches that seek to 'extrapolate' long videos from a few synthetic frames without applying appropriate inductive biases to the process. Second, we also outperform these methods in terms of long-range temporal consistency. The consistency is evident from the fact that our 'overall scene' remains the same throughout the 1024 frames. Whereas it gets destroyed or changes for the other methods. We attribute the improvement in consistency to DiT's (Peebles & Xie, 2023) strong self-attention prior, which provides inductive biases to our sampling algorithm. Third, our ability to synthesize much longer videos than the ones seen in training asserts our remediation of the widely prevalent leakage (Somepalli et al., 2023) and memorization (van den Burg & Williams, 2021) issues in generative models.

Figure 15 illustrates a similar analysis for the SkyTimelapse dataset (Zhang et al., 2020). Although the aforementioned benefits of our method are evident here as well, a few more interesting ones emerge. Specifically, we observe that LVDM (He et al., 2022) collapses to a mode of dark scenes across various iterations of unconditional sampling. Whereas, StyleGAN-V (Skorokhodov et al., 2022) performs comparably to our method in terms of per-frame quality. Yet, it is worse in terms of FVD due to the presence of 'looping artifacts', which we resolve in our method using 3D positional embeddings. Moreover, both these methods struggle in terms of variability. LVDM struggles with viewing angle and lighting variations across and within its synthetic videos. However, StyleGAN-V struggles only in per-frame variability within different synthetic videos. Our method performs better on both of these fronts.

## C.2 Different sampling settings

We dedicate this section to examining the interplay between the various variables associated with our Grid-based autoregressive sampling scheme. In figures 16, 17, and 18 we present the first five iterations of our Grid-based Autoregressive sampling algorithm in its $\{K = 2$, one-row control signal$\}$, $\{K = 4$, three-row control signal$\}$, and $\{K = 8$, four-row control signal$\}$ settings on the Taichi dataset, respectively for step 1 of our sampling scheme. In all these experiments, we interpolated $K/2$ grid elements in step 2 of the sampling scheme. The figures conform to the findings of section 4.3 of the paper, wherein we observe a tradeoff between the amount of temporal signal and spatial details a setting has to offer.

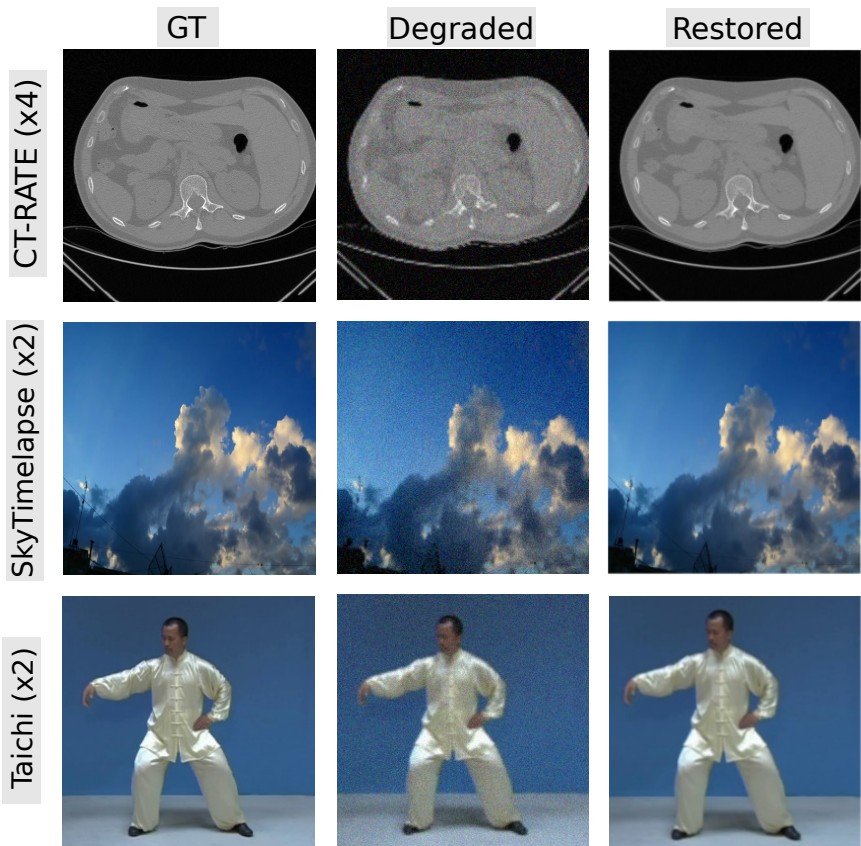

Figure 11: An overview of the surrogate restoration task performed by our method's stage 2 (②) to perform individual frame refinement. (GT: Ground Truth).

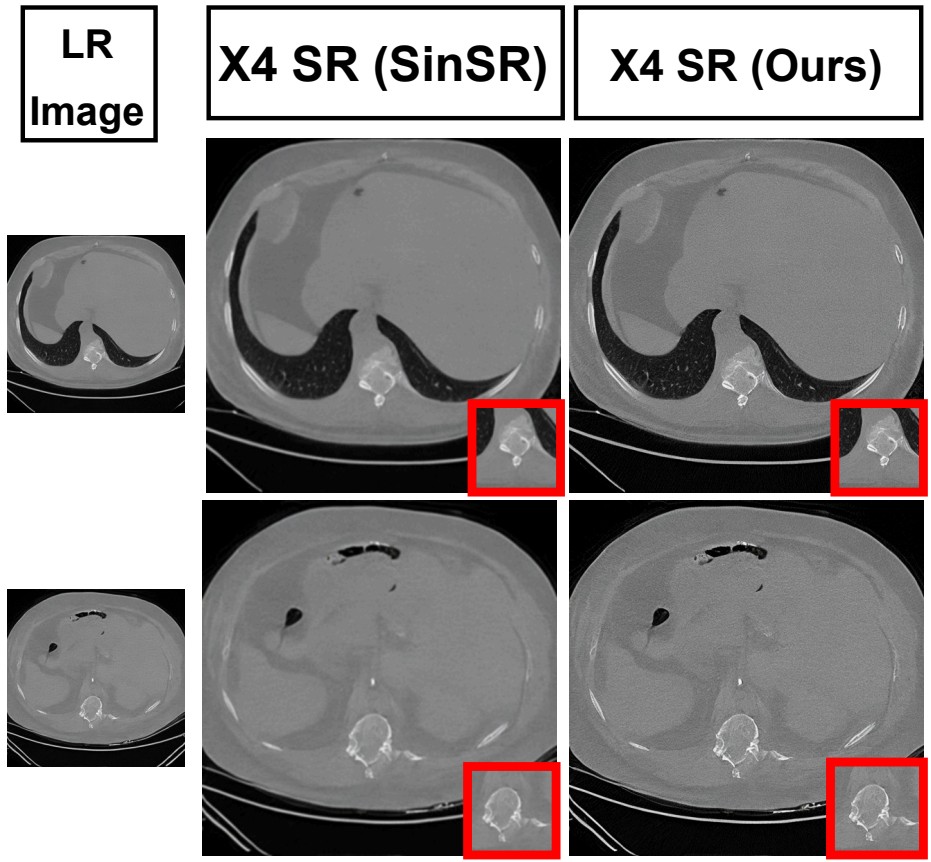

Figure 12: Qualitative comparison for the SR task on the CT-RATE dataset. Our method's stage 2 (②) qualitatively outperforms SinSR (finetuned) on the super-resolution and fine information addition task, especially in high-frequency regions, as highlighted by the inset.

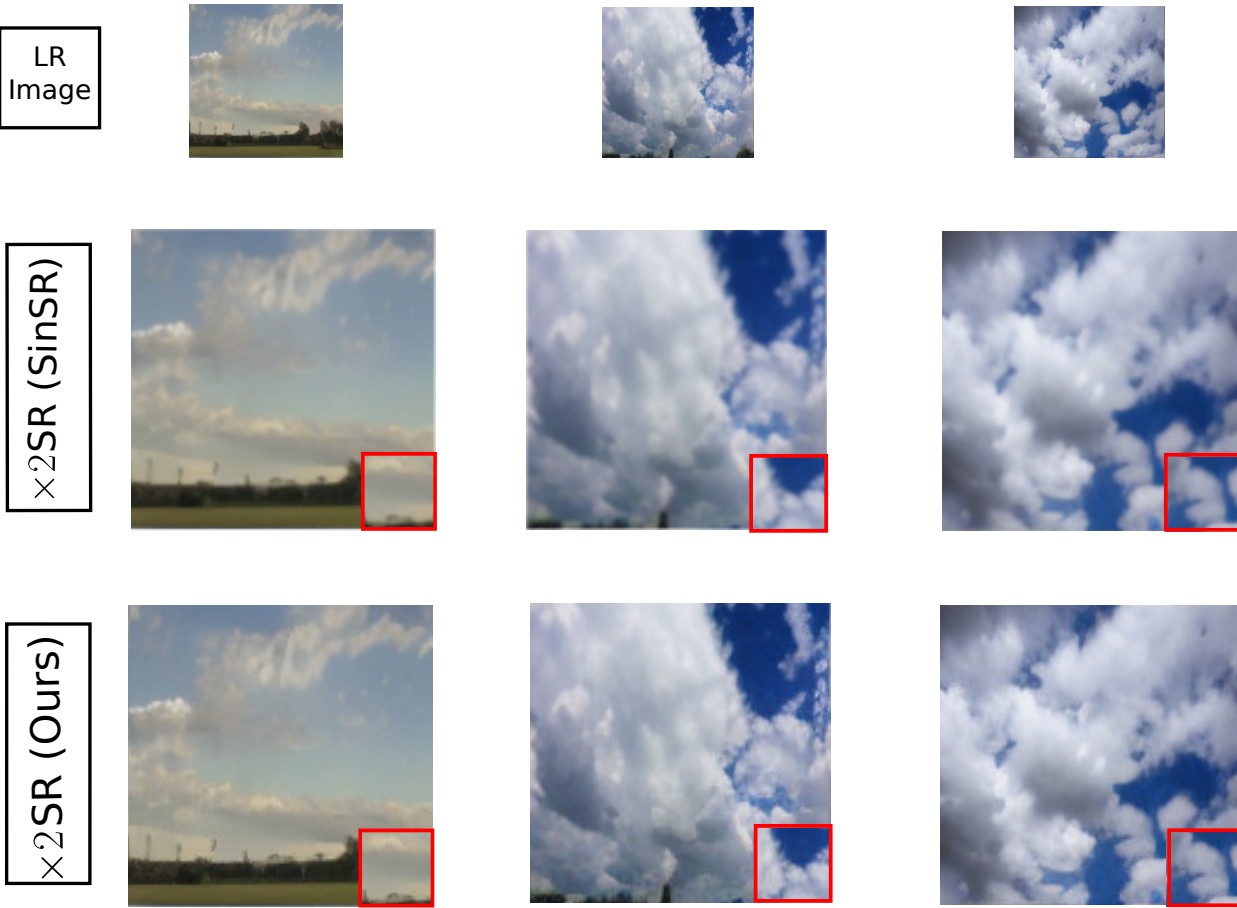

Figure 13: Qualitative comparison for the SR task on the SkyTimelapse dataset. Our method's stage 2 (②) qualitatively outperforms SinSR in refining individual frames, as highlighted by the inset.

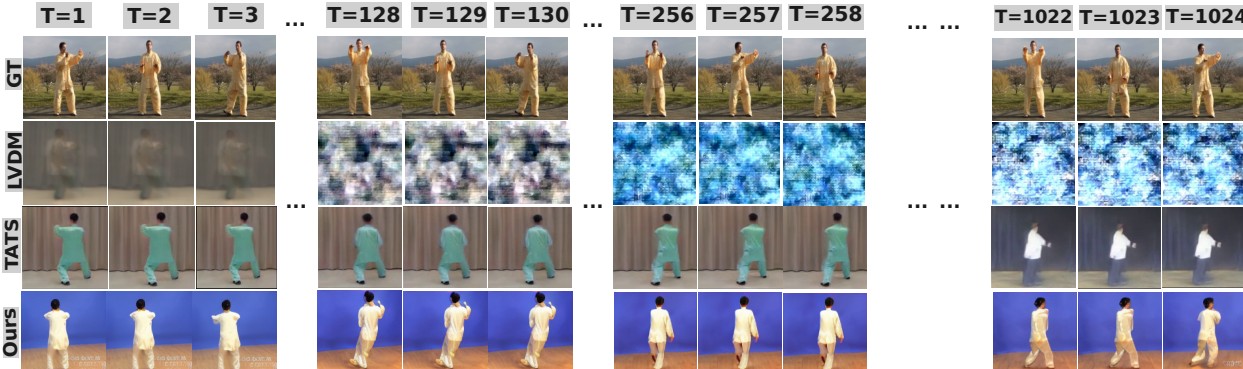

Figure 14: Qualitative comparisons for arbitrary length video synthesis at $256 \times 256$ resolution on the Taichi dataset. As presented in our Quantitative results in section 4.1 of the paper, our method offers: **(1) Superior perceptual quality** than the prior art, exemplified by the sharpness of details in our synthetic frames. **(2) Superior spatiotemporal consistency** than the prior art. There are no changes in the overall 'scene' or 'random jumps' even when the synthesis is extended to arbitrary lengths. **(3) Freedom from leakage and memorization issues** as it generates arbitrarily long videos despite being trained only on videos with $\leq 300$ frames. ($T$ : Frame indices. GT: Ground Truth. Please note that the ground truth video has been looped repeatedly to display 1024 frames, despite being only 293 frames long.)

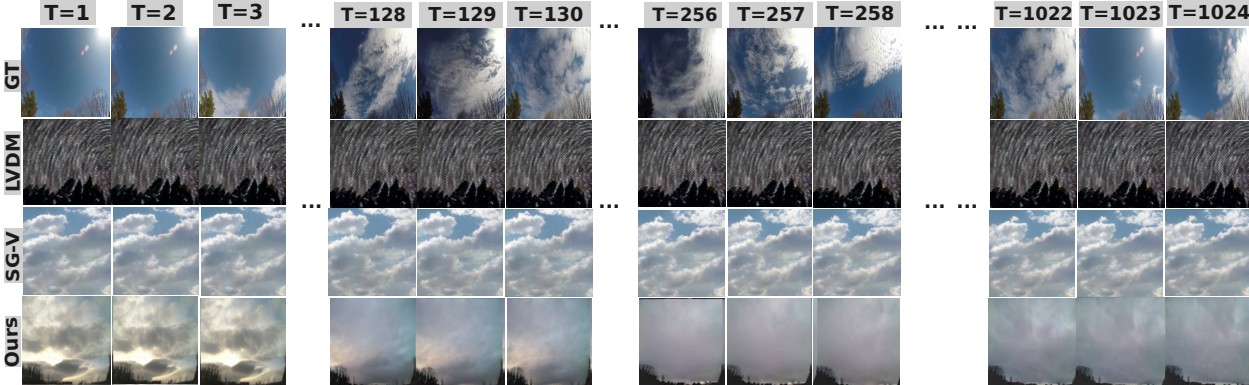

Figure 15: We compare our method with the SoTA for arbitrary length video synthesis at $256 \times 256$ resolution on the SkyTimelapse dataset qualitatively. Consequently, we observe that: **(1)** LVDM struggles in terms of photorealism of the synthetic frames and modeling motion across them. Moreover, it mostly generates scenes in low light with limited variability. **(2)** StyleGAN-V yields comparable per-frame quality but struggles in terms of variation in the scene lighting and motion across frames. We observed the 'looping artifacts' that we specifically mitigate by using 3D positional embeddings in our method. **(3)** Whereas our method generates high-quality spatiotemporally coherent videos with substantial diversity in lighting across different frames. ($T$ : Frame indices. GT: Ground Truth. SG-V: StyleGAN-V. Please note that the ground truth video has been looped repeatedly to show 1024 frames despite it being only 361 frames long.)

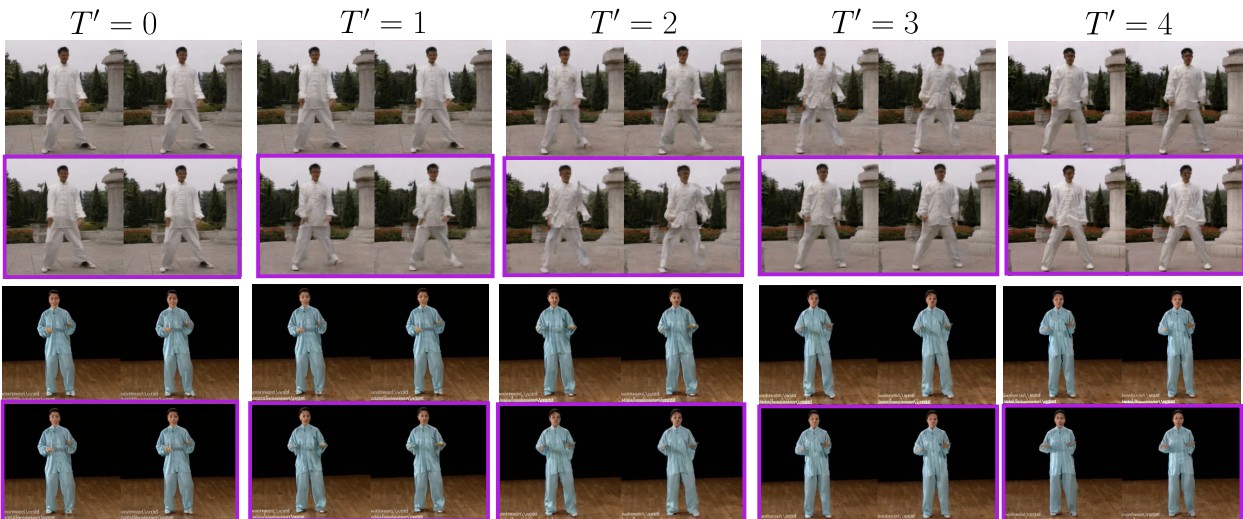

Figure 16: We perform Grid-based Autoregressive sampling for arbitrary length video generation in the $\{K = 2,$ one-row control signal$\}$ setting on the Taichi dataset. We observe that the setting sacrifices spatiotemporal consistency for slight gains in per-frame quality. (⬜: Sampling control signal.)

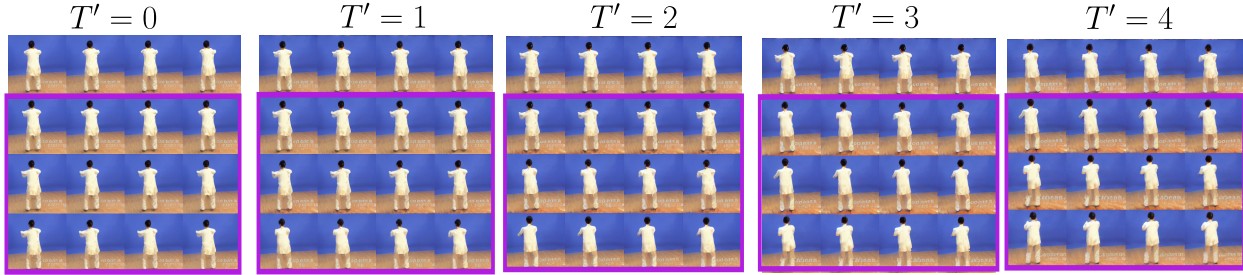

Figure 17: We perform Grid-based Autoregressive sampling for arbitrary length video generation in the $\{K = 4,$ three-row control signal$\}$ setting on the Taichi dataset. This setting is the sweet spot of the tradeoff between per-frame quality and long-range temporal consistency. (⬜: Sampling control signal.)

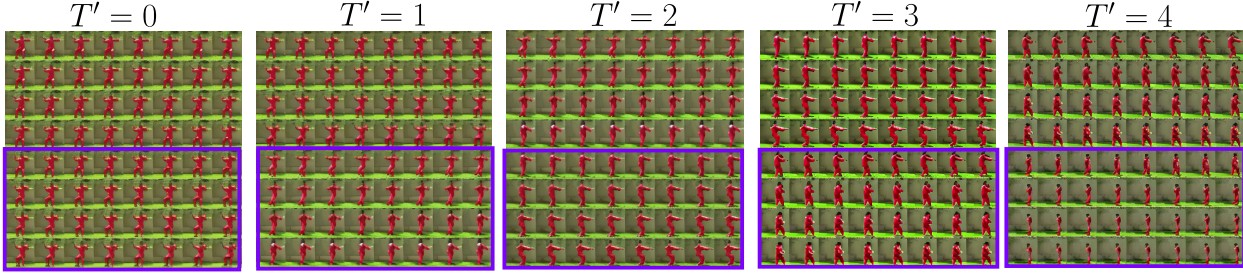

Figure 18: We perform Grid-based Autoregressive sampling for arbitrary length video generation in the $\{K = 8,$ four-row control signal$\}$ setting on the Taichi dataset. We observe that the setting sacrifices per-frame quality for slight gains in spatiotemporal consistency. (⬜: Sampling control signal.)

