# OpenReview forum: "GriDiT: Factorized Grid-Based Diffusion for Efficient Long Image Sequence Generation"
_TMLR — Accepted by TMLR_

### Review · Reviewer_sa38 · 2025-12-26

**Summary Of Contributions:**

The authors propose an efficient way to generate image sequences (i.e., videos or 3D volumes) using SOTA image generation approaches. Their approach consists of two stages: (1) synthesizing a grid image of low-resolution frames and (2) super-resolving the individual frames in the grid image. An unconditional Diffusion Transformer (DiT) is used for Stage 1, while a conditional DiT is used for Stage 2. They also propose a grid-based autoregressive sampling algorithm to generate arbitrarily-long image sequences. The grid-based autoregressive sampling works by recursively generating new grid images, and then these grid images are all super-resolved via Stage 2.

The authors provide extensive experimental results showing the superior perceptual quality and efficiency of their approach over existing baselines for the CT-RATE, SkyTimelapse, and Minecraft datasets. They also provide extensive ablation studies.

Strengths:
* The proposed method is sound.
* The manuscript is very well-written and easy to follow.
* The experimental results are quite convincing regarding the superior perceptual quality and efficiency of GriDiT over previous methods.
* The authors provide extensive ablation studies to justify their design choices.
* The authors are upfront about the limitations of the method (e.g., struggling with fine motion of small objects between frames).

I don’t see any weaknesses besides the minor ones already addressed by the authors. I just have a nitpicky suggestion to define abbreviations like “INR.”

**Audience:**

Yes

**Audience Explanation:**

The proposed method is relevant to people working on video/volume generation and/or reconstruction.

**Claims And Evidence:**

Yes

**Claims Explanation:**

The authors provide lots of experiments comparing to other baselines, as well as ablation studies.

**Requested Changes:**

None

---

> ### Author Response · Authors · 2026-01-13
> **Response to Reviewer sa38**
>
> We thank the reviewer for their in-depth review of our paper. Their overall positive assessment of our work comforts us in the quality of our contributions. We are glad they found our method sound, our experiments well-designed, and our manuscript well written.
>
> Keeping the reviewer's suggestion in mind, we have explicitly elaborated upon the standard abbreviations given below in our revised manuscript.
>
> * INRs: Implicit Neural Representations
> * GANs: Generative Adversarial Networks
> * CT: Computed Tomography
> * VAE: Variational Autoencoder
>
> We have highlighted these changes in blue font in sections 1, 2, and 3 of the revised manuscript.

---

### Review · Reviewer_iBBg · 2025-12-28

**Summary Of Contributions:**

The paper proposes a new method for efficient video / image sequence generation called GriDiT. The method adopts a two-stage approach of first generating a grid consisting of video frames / individual images and then upscaling them via super-resolution.

**Strengths**
- S1: The approach is intuitive and produces short-form videos of higher quality at significantly lower latency than competitive baselines.
- S2: The authors are generally thorough with their method description and experimental setup, and the paper is well written overall.

**Weaknesses**
- W1: The proposed stage-1 generation is hard to understand from Fig. 3 and Sec 3.2. The section mentions that 4 new frames are generated in each iteration of stage-1, and 8 frames are in-painted from the generated 4 frames in consecutive iterations T' and T'+1.
  - W1.1 Does this mean step-1 and step-2 are alternating? What's the rationale for such back and forth generation for 4 and 8 frames?
  - W1.2 Shouldn't native video FPS / resolution affect these values? Is there any relation between K and these 4, 8, 12 frames?
  - W1.3 The snowflake symbol is generally associated with frozen model / model layers in literature but aren't both stage-1 and 2 DiTs being trained fully?

- W2: One of my main concerns is the scope of some claims made -- the paper re-iterates throughout Sec 4 that the method allows modeling significant motion and/or fine-grained changes in long video sequences.
  - W2.1 Consider a few minutes long video at 30 FPS at Full HD resolution which is common for real-life use cases, is (1) generating a collage of sparsely sampled frames (without explicit causal modeling) and then (2) upscaling via super-resolution really the right approach for generating such videos? It doesn't seem intuitive as (1) is bound to lose spatiotemporal changes across frames and (2) is meant to upscale individual frames and not recover such information loss.
  - W2.2 Both the Taichi and Minecraft datasets are limited in terms of one or more of video length (1024x1024 frames), resolution (upto 256x256), slow-moving objects of interest or finite action space. While these are common benchmarks and the authors are in the right for using them for comparison, the fundamental limitations of their method and benchmarks and scope of improvement should be stated more clearly for future works.

**Audience:**

Yes

**Audience Explanation:**

Yes, both researchers and practitioners will find this paper useful due to the intuitive high-level method design which is also more performant than other competing methods.

**Broader Impact Concerns:**

None, broader impact concerns are described in A.11 section of the appendix.

**Claims And Evidence:**

Yes

**Claims Explanation:**

- The overall claim of the proposed two-stage GriDiT method being effective for video / image sequence generation is met through generation quality and latency benchmarking experiments against recent competitor baselines.
- However, some other claims such as W2 above seem a bit misleading and should either be worded properly so that the scope and limitations of the method are clear or accompanied with experiments to better justify these claims.

**Requested Changes:**

Please check the weaknesses section above. In my opinion, W2 is critical for paper acceptance and W1 will strengthen the work and make it easier for the readers to understand and follow-up on.

---

> ### Author Response · Authors · 2026-01-13
> **Response to Reviewer iBBg - Part I**
>
> We are grateful to the reviewer for their painstaking efforts in providing a detailed review to improve the paper. We are glad they found our approach intuitive and novel, and our manuscript thorough and well-written. We appreciate the detailed feedback and address (A) the concerns (W) one by one below.
>
> * **W1**: The proposed stage-1 generation is hard to understand from Fig. 3 and Sec 3.2. The section mentions that 4 new frames are generated in each iteration of stage-1, and 8 frames are in-painted from the generated 4 frames in consecutive iterations T' and T'+1.
>     * **W1.1**: Does this mean step-1 and step-2 are alternating? What's the rationale for such back and forth generation for 4 and 8 frames?
>        * **A1.1**: No, we execute step 2 of our sampling scheme _after_ step 1 and not in an alternating fashion. In step 1, we simply set $N=(L+4)/12$ where $L$ is the required number of output frames post conclusion of both the steps and $N$ is the required number of step 1 iterations required to get $L$ output frames after step 2. We then proceed with step 2 to interpolate 8 new frames between the 4 latest/novel frames obtained in each step 1 iteration to get $L$ output frames. Moreover, the motivation for adding a second stage of interpolation-based sampling lies in the need to reduce observed discontinuities and enhance temporal stability after sampling with step 1 only. The newly interpolated grid elements (later super-resolved to frames of required dimensions) help in two ways. First, they reduce subtle jumps or temporal instability artifacts in the synthetic sequence. We are unable to report an ablation quantifying this effect, as metrics such as FVD and average L1 frame distance are not sensitive enough to capture the subtle improvement. Second, since we generate 8 new frames in each step 2 iteration (T'') instead of 4 in each step 1 iteration (T'), the total iterations required for a given sequence length decrease, yielding further gains in sampling time. These clarifications are added to Section 3.2 and Algorithm 1 in the revised manuscript.
>     * **W1.2**: Shouldn't native video FPS / resolution affect these values? Is there any relation between K and these 4, 8, 12 frames?
>         * **A1.2**: We would like to bring forth the following points in this regard:
>             * One of the central pillars of our contributions in this work is the ability to _repurpose_ a 2D (image) generative model (DiT [1]) as a 3D (image-sequence) generative model without any architectural changes by treating image-sequences as grids of subsampled frames/slices in stage-1. Consequently, we chose the DiT-XL model in its $512\times512$ dimensional variant as the base model in stage 1 and worked with different values of $K$ depending upon the required output resolution. Essentially, although fixing DiT-XL-512 as the base model always sets the grid-element resolution to $512/K$ in stage 1, the upscaling factor in stage 2 is, in fact, **varied** depending on the required output resolution.
>            * Although we do not explicitly factor in the native (capture) frame rate of the ground truth videos in our modeling scheme, and the formulation works well for the examined datasets, we acknowledge that this might be an issue for other datasets with finer motion. In those cases, adding the desired FPS as a conditioning signal to our stage-1 model and to both steps of the sampling algorithm could alleviate the issue. We have added the same to the _Limitations and Future Work_ section in the revised manuscript.
>             *  With respect to our design choices: (1) The setting $K=4$ was arrived at after a careful analysis of the _temporal versus spatial context tradeoff, given a grid size_ in our ablation study on grid size in section 4.3 of the paper. (2) In step 1 of the Grid-based Autoregressive Sampling Algorithm, we chose 12 previously generated grid elements as the control (conditioning) signal and inpainted 4 new frames in each sampling iteration. The reasons for this choice were twofold. First, we wanted to provide the maximum possible conditioning from the previously generated grid elements to inpaint new grid elements. Second, we wanted to generate enough new frames in each sampling iteration such that our sampling efficiency does not get compromised. (3) Similarly, in each step 2 iteration $T''$, we use the maximum possible, i.e., 4 new grid-elements from the previous ($T'$) and the next ($T'+1$) step 1 iterations, respectively, to inpaint the remaining 8 elements in a $4\times4$ grid. (4) We chose the design settings (2) and (3) empirically. Later, we found that these choices worked better qualitatively than other candidate design choices in our initial experiments. Hence, we performed all our reported experiments according to these choices. Some of the candidate design choices we studied included: (a) inpainting 12 new grid elements with only 4 element conditioning in step 1, and (b) sampling with step 1-only.

---

> ### Author Response · Authors · 2026-01-13
> **Response to Reviewer iBBg - Part II**
>
> * **W1.3**: The snowflake symbol is generally associated with frozen model / model layers in literature but aren't both stage-1 and 2 DiTs being trained fully?
>    * **A1.3**: The snowflake symbol in Figure 2 signifies that the VAE elements are frozen during training. In Figures 3 and 4, this signifies that the model is frozen during inference. We have updated all relevant figure captions to reflect the same in the revised manuscript.
>
>     Finally, we would like to point the reviewer to Algorithms 1 and 2 in Appendix A.5 for a formal explanation of the Grid-based Autoregressive Sampling algorithm.
>
> * **W2**: One of my main concerns is the scope of some claims made -- the paper re-iterates throughout Sec 4 that the method allows modeling significant motion and/or fine-grained changes in long video sequences.
>     * **W2.1**: Consider a few minutes long video at 30 FPS at Full HD resolution which is common for real-life use cases, is (1) generating a collage of sparsely sampled frames (without explicit causal modeling) and then (2) upscaling via super-resolution really the right approach for generating such videos? It doesn't seem intuitive as (1) is bound to lose spatiotemporal changes across frames and (2) is meant to upscale individual frames and not recover such information loss.
>         * **A2.1**:  We understand the reviewer's standpoint in this regard and would like to submit the following for their kind consideration:
>             * Downsampling is a common and nearly essential practice across all categories of Diffusion Models, especially Latent Video Diffusion Models, to make the involved computations tractable. Yet, these models yield State-of-the-Art (SoTA) performance. The key difference between such methods and ours is that we upsample the frames independently, whereas such methods upsample entire video representations at once. However, such a formulation is not suitable for generating arbitrarily long image sequences, as it is infeasible to: (1) decode/super-resolve/upsample arbitrarily long sequences all at once, or (2) have a single deep feature representation for arbitrarily long sequences and super-resolve it. Moreover, our formulation yields high-quality results on all the examined datasets.
>             * As shown in our section on "Mechanistic Insights, " in our formulation, GriDiT's self-attention mechanism, paired with its 3D positional embeddings, is responsible for learning the temporal context and relative variation per frame between grid elements. Whereas, the role of upsampling in stage 2 is to restore and add spatial details only. Hence, our method works well despite stage 2's contributions being purely spatial in nature.
>             * We contrast our approach with a 3D VAE-based approach. Wherein, a fixed-length video is downsampled to a latent representation that is employed in the diffusion process. High-resolution videos are then decoded from the latent after the diffusion process concludes. We report these results in "Naive Baseline II" in Appendix A.11 of the paper. Therein, our `Grid-based formulation + DiT` approach, which upsamples individual frames independently, outperforms a `3D VAE + DiT` approach even for short (16) length videos, which upsamples all 16 frames from a unified representation. Thereby highlighting the modeling capabilities of our method. We used the 3D VAE employed by CogVideoX [2] for this experiment.
>             * Despite the above, we do agree that we haven't been able to thoroughly validate our approach for full HD videos with very fast-moving objects that are unobservable at a grid element's resolution. In light of this and taking the reviewer's advice into account, we have reworded our claims and limitations to reflect the same clearly. We elaborate on the specific modifications we made to our revised manuscript in this regard, in **A2.2** below.

---

> ### Author Response · Authors · 2026-01-13
> **Response to Reviewer iBBg - Part III**
>
> * **W2.2**: Both the Taichi and Minecraft datasets are limited in terms of one or more of video length (1024x1024 frames), resolution (upto 256x256), slow-moving objects of interest or finite action space. While these are common benchmarks and the authors are in the right for using them for comparison, the fundamental limitations of their method and benchmarks and scope of improvement should be stated more clearly for future works.
>   * **A2.2**:  We agree with the reviewer's take on our claims and have reworded them in the revised manuscript to reflect our experimental findings and limitations more clearly. We have made those modifications to sections 4.2 (our results) and 5 (limitations and future work) of the revised manuscript. To be specific, we have stated in our results that "Our results on datasets with high variability, such as Taichi and Minecraft, underscore the efficacy of our method in sampling arbitrarily long synthetic sequences from most real-world datasets, provided their frame resolution, motion content, and degree of variability lie within the bounds of those quantities in our studied datasets." We have also added the following to our limitations and future work: "It is important to further validate our method's performance on datasets with higher frame resolution and motion content than those of our studied benchmarks."
>
> We have highlighted all changes related to the reviewer's comments in violet font in the revised manuscript.
>
> #### References:
>
> [1] Peebles, William and Xie, Saining. "Scalable Diffusion Models with Transformers," ICCV 2023.
> [2] Yang et al. "CogVideoX: Text-to-Video Diffusion Models with An Expert Transformer," ICLR 2025.

---

> > ### Comment · Reviewer_iBBg · 2026-01-14
> > **Official Comment by reviewer iBBg**
> >
> > Thanks authors for following-up on the comments with appropriate clarifications and revised text. Most of my concerns have been addressed and I retain my overall positive review of the paper and recommend acceptance. Sharing a few of my remaining thoughts below:
> >
> > - A1.3: Thanks for the clarification, a minor comment is during inference frozen and snowflakes symbols are redundant as that's expected to be the case unless for test-time adaptation / training which isn't the focus of this paper.
> >
> > - A2.1: To clarify, my perspective in W2.1 is more from a conceptual point-of-view rather than comparison with existing state-of-the-art methods or ablations for GriDiT which the authors to their credit have performed extensively. Nonetheless, this discussion is not necessarily in the scope of the paper and the revised text is more forthcoming with the scope of future work.
> >
> > - A2.2: Thanks for revising the paper, however page L11 still mentions "significant motion across frames and fine-grained details" for Taichi dataset which I'd also recommend updating in accordance with W2.2 and the discussion thereafter.

---

> > > ### Author Response · Authors · 2026-01-15
> > > **Thanks for the acceptance!**
> > >
> > > We thank the reviewer for recommending our work to be accepted. We address the remaining concerns below:
> > >
> > > * We have removed the snowflake symbol from Figures 3 and 4.
> > > * We have also reworded our analyses in the ablation study on Grid Size ($K$).
> > >
> > > These changes are highlighted in violet font color in the revised manuscript.

---

### Review · Reviewer_y8Vf · 2026-01-02

**Summary Of Contributions:**

This paper proposed GriDiT, which is a multiscale type of Diffusion Transformer (DiT) for efficient and long image-sequence generation based on factorized grids. The core idea is to represent a sequence of images as a 2D grid of subsampled frames, train a DiT on these grids at low resolution (Stage 1), and then apply a conditional DiT-based super-resolution model to refine each frame independently at high resolution (Stage 2). A Grid-based Autoregressive Sampling procedure then enables generating sequences of arbitrary length by iteratively inpainting new grid rows conditioned on previously generated rows, followed by temporal interpolation between successive grids.

**Additional Comments:**

Overall, the reviewer thinks that this paper studied an important topic, but it needs some refinement (especially a more comprehensive review on related work) before being considered for TMLR.

References:

[1] Zhang, Shumao, Pengchuan Zhang, and Thomas Y. Hou. "Multiscale invertible generative networks for high-dimensional Bayesian inference." International Conference on Machine Learning. PMLR, 2021.

[2] Guth, Florentin, et al. "Wavelet score-based generative modeling." Advances in neural information processing systems 35 (2022): 478-491.

[3] Kadkhodaie, Zahra, et al. "Learning multi-scale local conditional probability models of images." arXiv preprint arXiv:2303.02984 (2023).

[4] Guth, Florentin, et al. "Conditionally strongly log-concave generative models." International Conference on Machine Learning. PMLR, 2023.

[5] Mei, Song. "U-nets as belief propagation: Efficient classification, denoising, and diffusion in generative hierarchical models." arXiv preprint arXiv:2404.18444 (2024).

**Audience:**

Yes

**Audience Explanation:**

To the best of the reviewer's knowledge, the topic studied in this paper (efficient long video and image-sequence generation with diffusion transformers) is an active research area, which also connects to ongoing efforts on factorized generation and autoregressive-based diffusion models. Therefore, the reviewer does think that the TMLR community will find the results presented in this paper interesting.

**Claims And Evidence:**

Yes

**Claims Explanation:**

The reviewer thinks that the authors have provided sufficient details to support the claims made in the manuscript, such as empirical results (plots in Figure 6 and Tables 1–3 & 7, qualitative sequences in Figures. 5, 14, 15 and ablation studies in Tables 4,6,8) and limitations (Section 5).

**Requested Changes:**

Though the empirical results presented in this paper are quite strong, the reviewer thinks that the quality of the manuscript can be further improved by including a paragraph to discuss the relation between the proposed methodology and existing work (An incomplete list: [1,2,3,4,5]) on the applications of multiscale methods in generative modeling.

---

> ### Author Response · Authors · 2026-01-13
> **Response to Reviewer y8Vf**
>
> We appreciate the reviewer's thorough analysis of our work. We thank them for finding our work interesting and having an overall positive opinion about it. We are grateful for their thoughtful suggestion of incorporating additional Prior Work on "Applications of multi-scale methods in generative models" to ensure completeness and improve the paper. We have added the same to the revised manuscript. We have highlighted the added content in red font in section 2 and Appendix A.1 of the revised manuscript.

---

### Decision · Action_Editor_EiD1 · 2026-02-14

**Recommendation:** Accept as is

**Audience:**

Yes

**Audience Explanation:**

This paper addresses efficient and scalable image-sequence and video generation using diffusion transformers, which is an active and rapidly evolving area within the machine learning community. The proposed grid-based factorization and autoregressive sampling framework contribute to ongoing discussions on multiscale modeling, long-sequence generation, and computational efficiency in generative models. Researchers working on diffusion models, video generation, generative transformers, and efficient architectures would find the methodological design and empirical analysis informative. In addition, practitioners interested in latency–quality trade-offs for generative systems may also benefit from the findings.

**Claims And Evidence:**

Yes

**Claims Explanation:**

All three reviewers ultimately support acceptance. They agree that GriDiT presents a technically sound and well-executed approach to efficient image-sequence and video generation. The two-stage grid-based formulation is clearly motivated, the method is carefully implemented, and the experimental evaluation is extensive. In particular, the empirical results demonstrate strong performance in perceptual quality and inference efficiency compared to competitive baselines, and the manuscript is generally clear and well structured.

During review, several substantive concerns were raised regarding the clarity of the grid-based autoregressive sampling procedure, the scope of claims about long-sequence and fine-motion modeling, and the positioning of the work relative to prior multiscale and factorized generative approaches. The authors responded constructively, provided detailed clarifications, refined the scope of their claims, expanded the related work discussion, and updated figures and presentation details accordingly. The reviewers confirmed that these revisions addressed their concerns and maintained their positive recommendations.

While the work builds on existing coarse-to-fine and multiscale ideas in generative modeling, the specific grid-based factorization and its integration with diffusion transformers for long-sequence generation represent a meaningful and practically relevant contribution. The paper will be of interest to researchers working on diffusion models, video generation, and efficient generative architectures.

Based on the overall technical quality, clarity after revision, and consistent reviewer support, I recommend acceptance.

---

> ### Author Response · Authors · 2026-03-19
> **Thanks for the acceptance!**
>
> We thank the action editor and reviewers for helping us improve the paper throughout the feedback period. We believe that the feedback contributed to enhancing the clarity and soundness of the paper. We have updated the camera ready version. We have incorporated the responses to the reviewers in the text.